

# Modelling Cold Firn Evolution at Colle Gnifetti, Swiss/Italian Alps

Marcus Gastaldello[1], Enrico Mattea[1], Martin Hoelzle[1], and Horst Machguth[1]

[1]Department of Geosciences, University of Fribourg, Fribourg, Switzerland

**Correspondence:** Marcus Gastaldello (marcus.gastaldello@unifr.ch)

**Abstract.** The existence of cold firn within the European Alps provides an invaluable source of paleo-climatic data with the capability to reveal the nature of anthropogenic forcing in Western Europe over the preceding centuries. Unfortunately, continued atmospheric warming has initiated the thermal degradation of cold firn to that of a temperate firn facie, where infiltrating meltwater compromises this vital archive. However, there is currently limited knowledge regarding the physical transition of firn between these different thermal regimes. We present the application of a modified version of the spatially distributed Coupled Snow and Ice Model in Python (COSIPY) to the high-altitude glacierised saddle of Colle Gnifetti of the Monte Rosa massif, Swiss/Italian Alps. Forced by an extensively quality-checked meteorological time series from the Capanna Margherita (4,560 m a.s.l.), with a distributed accumulation model to represent the prevalent on-site wind scouring patterns, the evolution of the cold firn's thermal regime is investigated between 2003 and 2023. Our results show a prolongation of previously identified trends of increasing surface melt at a rate of 0.54 cm w.e. $yr^{-2}$ – representing a doubling over the 21 year period. This influx of additional meltwater and the resulting latent heat release from refreezing at depth, drives englacial warming at a rate of 0.056 °C $yr^{-1}$, comparable to in-situ measurements. Since 1991, a measured warming of 1.5 °C (0.046 °C $yr^{-1}$) has been observed at 20 m depth with a marked rotation in the temperature gradient in the uppermost 30 metres of the glacier – also partially reproduced by our model. In lower altitude regions ($\sim$ 4,300 m a.s.l.), simulated warming is much greater than the local rate of atmospheric warming resulting in a rapid transition from cold to temperate firn – potentially indicative of future conditions at Colle Gnifetti. However, the simulation is very sensitive to changes to the model's parameterisation in this area and validation data is scarce. We also reveal how small changes to the model spin-up have a major influence on the evolution of the modelled thermal regime.

## 1 Introduction

Glaciers can be differentiated with respect to their thermal characteristics: those that exist permanently at negative temperatures below the depth of seasonal variation being referred to as 'cold', whilst those maintaining a temperature at the pressure melting point of ice are known as 'temperate'. The majority of glaciers within the European Alps exist in a fully temperate state, however a few areas (particularly high altitude regions above 3,500 m a.s.l) enable the formation of cold firn and ice, leading these glaciers to attain a hybrid or polythermal regime (Suter and Hoelzle, 2002; Blatter and Hutter, 1991). The accumulation region of these glaciers can similarly be classified into zones of differing thermal regimes known as firn facies according to Shumskii (1955, 1964):



    - the (cold) re-crystallisation zone, where the surface temperature never reaches the melting point and as such no surface melt occurs.

    - the (cold) re-crystallisation-infiltration zone, where melt occurs but meltwater percolation is limited to the uppermost snow layers of the current accumulation year and does not penetrate into the firn layers beneath.

    - the cold infiltration zone, where meltwater can percolate through several layers of annual firn but is either internally retained or refrozen.

    - the temperate firn zone, where englacial temperatures at all depths tend to the melting point and meltwater can fully percolate through the firn.

Within the re-crystallisation and re-crystallisation-infiltration facies, the depositional stratigraphy of accumulated snow layers is not compromised by a significant infiltration of surface meltwater (Shumskii, 1964; Hoelzle et al., 2011). The analysis of particles within these distinguishable annual layers of cold firn can reveal the historic variability in atmospheric composition and the Earth's climate (Konrad, 2011; Licciulli et al., 2020). In Western Europe, the retrieval of such ice cores have been indispensable due to their close proximity to a principal source of historic anthropogenic emissions (Jenk et al., 2009; Legrand et al., 2013). Unfortunately, climate change has initiated the thermal degradation of existing cold firn areas where the infiltration of meltwater to increasing depths endangers the future longevity of these archives (Gabrielli et al., 2010; Hoelzle et al., 2011). Ultimately, a gradual transition of cold to temperate firn from areas of low to high elevation is anticipated over time (Lüthi and Funk, 2001; Vincent et al., 2007). However, the monitoring of englacial temperature changes can itself be a vital indicator of a cold or polythermal glacier's response to current atmospheric warming (Hoelzle et al., 2011). Unlike traditional glaciological measurements that focus on mass losses in the ablation zone, cold firn areas do not exhibit significant mass changes due to the retention and refreezing of meltwater; the additional energy instead manifests as an increase in englacial temperature (Haeberli and Beniston, 1998; Vincent et al., 2020).

Colle Gnifetti (CG), a high-altitude glacierised saddle in the Monte Rosa massif (Fig 1), has been at the forefront of alpine cold firn research for half a century, possessing an extensive archive of climatological measurements. Unique to any other cold firn site in Central Europe, the extremely low flow and accumulation rates at CG (minimum of 15 cm w.e. yr$^{-1}$ (Bohleber et al., 2013)), glacier thickness (average 100 m (Haeberli et al., 1988)) and favourable topographic setting, enable ice core dating into the millennial timescale up to the late Pleistocene (19.6 kyr BP) (Jenk et al., 2009; Wagenbach et al., 2012). Englacial temperatures have been subject to investigation at CG since 1977. Haeberli and Funk (1991) concluded that observed englacial temperatures derived from borehole measurements in 1982 remained near steady-state conditions with a limited influence of meltwater-refreezing. However, subsequent measurements in 1995 revealed inflexions in the thermal profile at shallow depth and were interpreted by Lüthi and Funk (2001) to indicate destabilisation and the onset of firn warming. Thereafter, Hoelzle et al. (2011) published further evidence of englacial warming at CG noting that areas beneath the saddle on the Grenzgletscher had already undergone a facie transition from a re-crystallisation-infiltration to a cold infiltration zone due to enhanced melt-



water percolation.

Suter and Hoelzle (2002) conducted a large scale fieldwork campaign across the wider Monte Rosa massif in 1999, with extensive borehole measurements aiming to determine the spatial extent of existing cold firn. Suter and Hoelzle (2002) noted strong firn temperature variation across the region, brought about by changes in insolation, accumulation, slope and surface
aspect. Further research included the analysis of surface energy exchanges from a temporary monitoring station at Seserjoch and the application of a basic time-dependent firn-temperature model to estimate future englacial warming rates (Suter et al., 2001, 2004). Later, Buri (2013) applied the GEO-top hydrological model using meteorological data from the nearby Automatic Weather Station (AWS) at the Capanna Margherita (CM) (operational since 2002), however both these models were only applied on isolated point nodes as opposed to a fully distributed spatial domain.


The application of physical firn models to CG is constrained by the complexity of representing the variability of accumulation. The exposed saddle is highly susceptible to extreme wind scouring of the snow surface that can even periodically lead to years with net mass ablation (Lüthi and Funk, 2001; Wagenbach et al., 2012). However, this process can be counteracted by the melt-induced consolidation of the snow-pack in the presence of strong insolation (Alean et al., 1983). Thus, snow accumulation
has a strong summer bias and adheres to the topographically-derived insolation gradient across the saddle, favouring the slopes of the Zumsteinspitze (Licciulli et al., 2020). Mattea et al. (2021) applied the spatially distributed, physically-based Energy Balance Firn Model (EBFM) of van Pelt et al. (2012) to CG, using extensively quality-checked, high-resolution hourly meteorological data from the AWS CM between 2003 and 2018. Accumulation variability was modelled using a spatio-temporally variable precipitation input, as the single dimensional model lacked a representation of lateral mass transference processes.
Mattea et al. (2021) investigated the evolution of firn temperatures and the dynamics of surface melt; reporting a pattern of increasing annual surface melt production – despite high inter-annual variability – perceived to be driving englacial warming at CG.

Here, we present the direct continuation of their research: the application of an alternative coupled energy balance and
multi-layer subsurface firn model to CG, extending the simulation period by five years to investigate changes in the thermal regime up to the present day. We further examine the nature of both simulated and observed firn warming, exploring the strong influence of the model spin-up and parameterisation selection on the output results.

## 2 Input Data

### 2.1 Meteorological Data

The meteorological forcing for the model is predominantly sourced from the CM AWS, situated on the summit of the Signalkuppe at an altitude of 4,560 m a.s.l. Hourly instantaneous values for air temperature, barometric pressure, wind speed and direction and global radiation are available from mid-2002 to the present day. However, as a result of the extreme operating



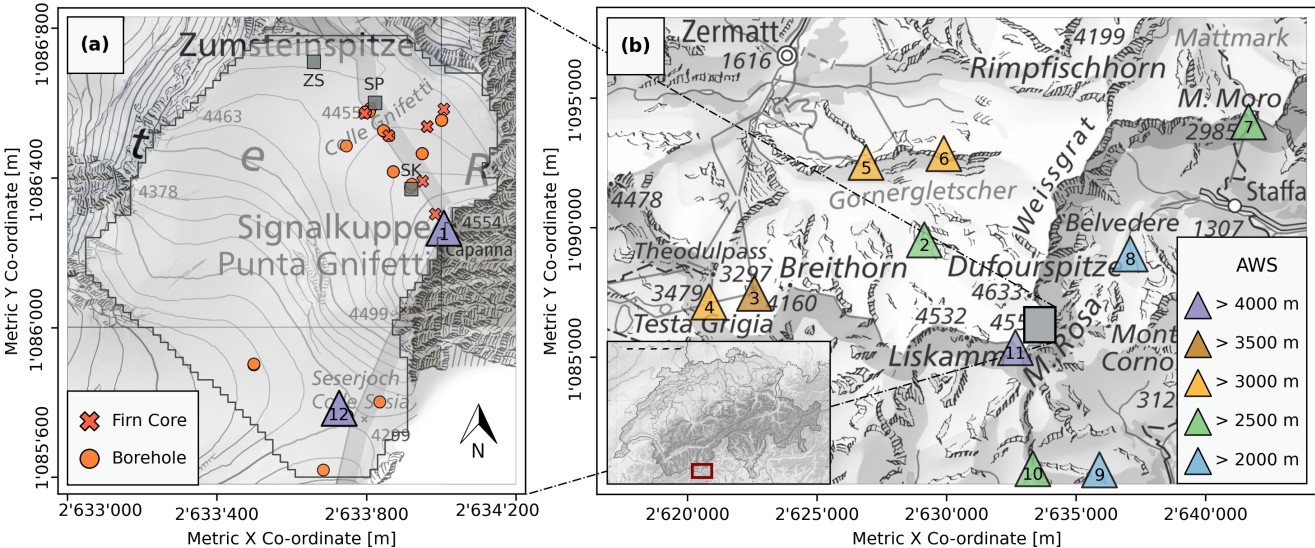

**Figure 1. (a)** Map of the study area at Colle Gnifetti showing key locations (ZS : Zumsteinspitze Slope, SP: Saddle Point, SK: Signalkuppe Slope) and borehole thermistor profiles extracted at the site. **(b)** Overview map of the Monte Rosa region showing the location of the study area and the meteorological stations (AWS) listed in Table 1. Spatial co-ordinates are defined by EPSG 2056 (Metric Swiss CH1903+/LV95). Topographic & orthographic map sources: Swisstopo (2017, 2022). Figure updated from Mattea et al. (2021).

conditions of the station, sensor malfunction and outages are of frequent occurrence (Martorina et al., 2003). Mattea et al. (2021) performed a comprehensive quality control process on the CM series between 2003 and 2018, reconstructing missing

data from 8 regional high altitude stations (Fig. 1, ID 4 - 10 in Tab. 1) using the technique of quantile mapping (Cannon et al., 2015; Feigenwinter et al., 2018). We acquired this dataset, then extended the meteorological time series by an additional 5 years up to the end of 2023 by closely replicating the processing steps detailed in Mattea et al. (2021) (Fig. 2). Due to data unavailability, we used the Kleines Matterhorn station (ID 3 in Tab. 1) in lieu of Plateau Rosa for air temperature, relative humidity and wind speed in our extension of the reconstructed series. However, their respective meteorological time series

were found to hold a strong correlation with each other.

Two of the required meteorological inputs for the model, relative humidity and fractional cloud cover, are not measured at the CM AWS and are therefore entirely reconstructed. Daytime fractional cloud cover is determined from incoming shortwave radiation measured at the CM AWS, whilst incoming longwave radiation at Stockhorn is used during the night. Relative

humidity is estimated as the arithmetic mean of the two highest altitude stations in the study region: Kleines Matterhorn and Stockhorn.





**Table 1.** Meteorological stations of the Monte Rosa Massif region used in this study. Positional co-ordinates adhere to the EPSG 2056 / Metric Swiss CH1903+/LV95 system. Table updated from Mattea et al. (2021).

| ID | Station Name | X [m] | Y [m] | Z [m a.s.l.] | Variables* | $D_{CG}$* [km] | Reference |
|---|---|---|---|---|---|---|---|
| 1 | Capanna Margherita | 2'634'007 | 1'086'266 | 4,560 | $T, S, P, V$ | 0.38 | ARPA Piemonte (2024) |
| 2 | Monte Rosa Plattje | 2'629'149 | 1'089'520 | 2,885 | $T, S, P, V, H$ | 5.53 | MeteoSwiss (2024) |
| 3 | Kleines Matterhorn | 2'622'588 | 1'087'457 | 3,883 | $T, V, R$ | 11.26 | DTN (2024) |
| 4 | Plateau Rosa | 2'620'839 | 1'087'122 | 3,488 | $T, P, V, R, N$ | 12.92 | MeteoAM (2020) |
| 5 | Gornergrat | 2'626'900 | 1'092'512 | 3,129 | $T, S, P, V, H$ | 9.21 | MeteoSwiss (2024) |
| 6 | Stockhorn | 2'629'900 | 1'092'850 | 3,415 | $T, S, L, P, V, H$ | 7.40 | Hoelzle et al. (2022) |
| 7 | Passo del Moro | 2'641'664 | 1'094'075 | 2,820 | $T, S, V, H, R$ | 11.01 | ARPA Piemonte (2024) |
| 8 | Macugnaga Refugio Zamboni | 2'637'094 | 1'088'977 | 2,075 | $T, S, V, H, R$ | 4.22 | ARPA Piemonte (2024) |
| 9 | Bocchetta della Pisse | 2'635'910 | 1'080'543 | 2,410 | $T, S, V, R$ | 6.41 | ARPA Piemonte (2024) |
| 10 | Passo dei Salati | 2'633'339 | 1'080'689 | 2,970 | $T, V, H$ | 5.89 | Visit Monte Rosa (2024) |
| 11 | Colle de Lys (temporary) | 2'632'665 | 1'085'360 | 4,236 | $T, S, V, H$ | 1.70 | Rossi et al. (2000a, b) |
| 12 | Seserjoch (temporary) | 2'633'727 | 1'085'785 | 4,292 | $T, S, L, P, V, H$ | 0.82 | Suter et al. (2004) |

(*Meteorological variable acronyms represent: $T$: air temperature | $S$: global shortwave radiation | $L$: longwave radiation | $P$: atmospheric pressure | $V$: wind speed | $H$: relative humidity | $R$: precipitation | $N$: cloud cover | $D_{CG}$ distance to the saddle point at Colle Gnifetti.)

## 2.2 Topographic Data

The model spatial grid is derived from a custom-built, 20 m resolution Digital Elevation Model (DEM). This was developed by Mattea et al. (2021) by amalgamating the publicly available SwissAlti3D and Piemonte DEMs – neither of which provide
full coverage due to the Swiss-Italian border bisecting the study area. The meteorological time series ($t$) is then projected onto this two dimensional spatial domain ($x$,$y$), creating a three dimensional model input file ($x$,$y$,$t$). Across the spatial domain, nodal air temperature and pressure are adjusted according to elevation differences using variable lapse rates. The remaining meteorological variables (wind speed, relative humidity and cloud cover) are assumed constant with elevation.

## 2.3 Subsurface Data

An extensive archive of subsurface data exists at CG that enables model validation. For evaluating the thermal regime, we used a total of 31 temperature profiles extracted from 18 boreholes during the simulation temporal range of 2003 - 2023. These profiles, detailed in Hoelzle et al. (2011) and GLAMOS (2020, 2022), are mostly concentrated in the vicinity of the saddle point and on the Signalkuppe flank of CG, but a few are located in the firn facie transition areas around the lower altitude Seserjoch (Fig. 1a). It is important to note that not all measurements are considered valid; some taken shortly after steam
drilling were discarded from our validation set as some of their thermistor readings were assessed to not be fully equilibrated. Stratigraphic data derived from firn cores obtained at CG (Fig. 1a) (Lier, 2018; Licciulli et al., 2020; Mattea et al., 2021) was





**Figure 2. (a)** The original quality checked and gap filled meteorological time series for the Capanna Margherita produced by Mattea et al. (2021) (2003-2018) and **(b)** the 5 year extension produced for this study (2019-2023). Hourly values are displayed with monthly means overlain as black lines; monthly cumulative values replace hourly values for the precipitation coefficient. The pie charts show the proportion of meteorological data in our extension directly sourced from CM AWS compared to that requiring reconstruction, due to being discarded during the quality control processes or missing due to sensor outages.





**Table 2.** Parameterisations selected for model implementation at CG compared against the default methods used in the baseline version 1.4 of the COSIPY model and the methods used by Mattea et al. (2021) with the Energy Balance Firn Model (EBFM) of van Pelt et al. (2012).

| Parameterisation | Selected Method | COSIPY Baseline v1.4 | Mattea et al. (2021) Method |
|---|---|---|---|
| Turbulent fluxes | Essery and Etchevers (2004)* | Foken (2008); Stull (1988) | Essery and Etchevers (2004) |
| Albedo decay | Bougamont et al. (2005)* | Oerlemans and Knap (1998) | Bougamont et al. (2005) |
| Fresh snow density | (constant) | Vionnet et al. (2012) | (constant) |
| Firn densification | Ligtenberg et al. (2011)* | Anderson (1976); Boone (2009) | Ligtenberg et al. (2011) |
| Penetrating radiation | Bintanja et al. (1995) | Bintanja et al. (1995) | N/A |
| Surface roughness | (constant) | Mölg et al. (2012) | (constant) |
| Thermal conductivity | Calonne et al. (2019)* | Bulk-volumetric | Sturm et al. (1997) |
| Specific heat capacity | Yen (1981)* | Bulk-volumetric | Yen (1981) |
| SEB numerical solver | SLSQP (amended) | L-BFGS-B, SLSQP or Newton | SLSQP |
| Sub-surface re-meshing | Fixed Lagrangian* | Adaptive Lagrangian or Logarithmic | Fixed Lagrangian |

(*New parameterisations added to the baseline version 1.4 of the COSIPY model.)

also used for the validation of simulated density profiles. Lastly, we installed a pair of continuously logging thermistor chains near the saddle point to measure changes in the firn's thermal regime at a 15 minute resolution during the summer 2022 melt season.

## 3 The Coupled Snow and Ice Model in Python

The Coupled Snow and Ice Model in Python (COSIPY), developed by Sauter et al. (2020), combines a skin-layer Surface Energy Balance (SEB) with a one dimensional, multi-layer subsurface model (Cryo Tools, 2022). In order to obtain representative results for CG, we had to extensively alter the baseline version 1.4 of COSIPY – often substituting parameterisations for those used in the EBFM (Tab. 2). This section outlines the model structure, detailing any major modifications we enacted for our implementation at CG. The baseline simulation results, alongside a discussion regarding the implications of these changes, are provided in Appendix A.

### 3.1 Surface Model

Driven by the meteorological forcing, the surface energy fluxes are evaluated at an infinitesimal skin layer to ascertain the surface temperature ($T_s$). Based on energy conservation,

$$\text{SW}_{\text{net}} \pm \text{LW}_{\text{net}} \pm Q_{\text{sensible}} \pm Q_{\text{latent}} \pm Q_{\text{subsurface}} = Q_{\text{melt}} \tag{1}$$





where $SW_{net}$ is the net shortwave flux, $LW_{net}$ is the net longwave flux, $Q_{sensible}$ and $Q_{latent}$ are the turbulent fluxes for sensible and latent exchange respectively and $Q_{subsurface}$ is the subsurface heat conduction flux. The surface temperature is physically constrained to 0 °C, therefore excess energy is apportioned to melt ($Q_{melt}$) should this condition arise.

### 3.1.1 Radiative Fluxes

The incident shortwave radiation for a given spatial node $(x,y)$ is modelled after Mölg et al. (2009):

$$SW_{in} = TOA\, \Lambda(x,y)\, \tau_{rg}\, \tau_{w}\, \tau_{a}\, \tau_{cl} \tag{2}$$

where TOA is the unattenuated top-of-atmosphere radiation and $\Lambda$ is a correction factor for angle of incidence and topographic shading. The coefficients of atmospheric transmissivity account for Rayleigh scattering and gaseous absorption ($\tau_{rg}$), water absorption ($\tau_{w}$) and the attenuation by aerosols ($\tau_{a}$) respectively. These are modelled after Kondratyev (1969), McDon-
ald (1960) and Houghton (1954) respectively. We modified the parameterisation for the attenuation of cloud cover ($\tau_{cl}$) after Greuell et al. (1997) to maintain parity with the meteorological input of the EBFM:

$$\tau_{cl}(N) = 1 - aN - bN^2 \tag{3}$$

where $a = 0.233$ and $b = 0.415$ are cloud transmissivity coefficients calibrated for high altitude alpine terrain and $N$ is the fractional cloud cover. The net shortwave radiation entering the SEB is calculated using a broadband, isotropic albedo ($\alpha$):

$$SW_{net} = SW_{in}(1 - \alpha) - SW_{pen} \tag{4}$$

where $SW_{in}$ is the input shortwave radiation and $SW_{pen}$ represents the penetrating radiation – an apportionment of the shortwave input that bypasses the SEB and directly warms the subsurface layers. This is modelled after Bintanja and van den Broeke (1995) where the absorbed radiation at depth $z$ is calculated as:

$$SW_{pen}(z) = \lambda_{abs}\, SW_{in}(1 - \alpha)\, e^{-z\beta} \tag{5}$$

where $\lambda_{abs}$ is the fraction of absorbed shortwave radiation (0.8 for ice; 0.9 for snow) and $\beta$ is the extinction coefficient (2.5 for ice; 17.1 for snow).

The evolution of the albedo is modelled after Oerlemans and Knap (1998) as an exponentially decreasing function of time $t$ since the last significant snowfall event, bounded by prescribed values for fresh snow ($\alpha_{fresh}$) and firn ($\alpha_{firn}$).

$$\alpha(t) = \alpha_{firn} + \left[ (\alpha_{fresh} - \alpha_{firn}) \cdot e^{-\frac{t}{t^*}} \right] \tag{6}$$





where $t^*$ is the characteristic decay timescale parameter. We modified this by adding the enhancement of Bougamont et al. (2005) that enables both a faster decay on a melting surface and slower metamorphism in cold conditions by introducing a dependence on surface temperature:

$$
t^*(T_s) =
\begin{cases}
t^*_{\text{wet}}, & T_{\text{s}} = 0\,^{\circ}\text{C} \\
t^*_{\text{dry}} + K\left[\max(T_{\text{s}}, T_{\text{max}, \, t^*})\right], & T_{\text{s}} < 0\,^{\circ}\text{C}
\end{cases}
\tag{7}
$$

where $t^*_{\text{wet}}$ and $t^*_{\text{dry}}$ are the decay timescales (d) for a melting and dry surface respectively, $K$ is a calibration parameter and $T_{\text{max}, \, t^*}$ is a temperature threshold for the decay timescale adjustment.

Net longwave radiation is calculated in accordance with the Stefan-Boltzmann law for grey body emission:

$$
\text{LW}_{\text{net}} = \text{LW}_{\text{in}} - \varepsilon_{\text{s}} \sigma T_{\text{s}}^4
\tag{8}
$$

where $\varepsilon_{\text{s}}$ is the surface emissivity and $\sigma$ is the Stefan-Boltzmann constant. In lieu of any input longwave radiation data at the site, the model parameterises this flux by substituting the air temperature and atmospheric emissivity ($\varepsilon_{\text{atm}}$) into the Stefan-Boltzmann law (Konzelmann et al., 1994).

$$
\varepsilon_{\text{atm}} = \varepsilon_{\text{cs}}(1 - N^2) + \varepsilon_{\text{clouds}} N^2
\tag{9}
$$

$$
\varepsilon_{\text{cs}} = 0.23 + c_{\text{emission}} \left(\frac{e_{\text{a}}}{T_{\text{a}}}\right)^{0.125}
\tag{10}
$$

where $\varepsilon_{\text{cs}}$ and $\varepsilon_{\text{clouds}}$ are the clear sky and cloud emissivities respectively and $c_{\text{emission}}$ is a calibration parameter.

### 3.1.2 Turbulent Fluxes

The turbulent fluxes are calculated according to a bulk aerodynamic approach after Foken (2008); Stull (1988):

$$
Q_{\text{sensible}} = \rho_{\text{a}} c_{\text{p,a}} C_{\text{h}} V_a (T_a - T_{\text{s}})
\tag{11}
$$

$$
Q_{\text{latent}} = \rho_{\text{a}} L_{\text{s,v}} C_{\text{h}} V_a (q_a - q_{\text{s}})
\tag{12}
$$

where $\rho_{\text{a}}$ is the dry air density (kg m$^{-3}$), $c_{\text{p,a}}$ is the specific heat of dry air under constant pressure (J kg$^{-1}$ K$^{-1}$), V is the wind speed (m s$^{-1}$), $L_{\text{s,v}}$ is the latent heat of sublimation or vaporisation (J kg$^{-1}$), $q$ is the specific humidity (kg kg$^{-1}$) and the $s$ and $a$ subscripts refer to the surface and the atmosphere at a measurement height of 2 metres respectively. We adapted the calculation of the turbulent exchange coefficient $C_{\text{h}}$ to follow the approach of Essery and Etchevers (2004):



$$C_{\mathrm{h}} = C_{\mathrm{hn}} \Psi_{Ri} \tag{13}$$

$$C_{\mathrm{hn}} = \kappa^2 \left[ \log \left( \frac{z_{\mathrm{a}}}{\mu} \right) \right]^{-2} \tag{14}$$

where $C_{\mathrm{hn}}$ is the value under neutral conditions, $\kappa$ is the von Karman constant and $\mu$ is the surface roughness. The stability function ($\Psi_{Ri}$), derived from the bulk Richardson number ($Ri_{\mathrm{b}}$), represents a correction for the stability of the atmospheric boundary layer.

$$\Psi_{Ri} = \begin{cases} (1 + 10 Ri_{\mathrm{b}})^{-1}, & Ri_{\mathrm{b}} \geq 0 \\ \left(1 - 10 Ri_{\mathrm{b}} \left(1 + 10\, C_{\mathrm{hn}} \frac{\sqrt{-Ri_{\mathrm{b}}}}{f_z}\right)^{-1}\right), & Ri_b < 0 \end{cases} \tag{15}$$

$$Ri_{\mathrm{b}} = \frac{g z_{\mathrm{a}}}{V_{\mathrm{a}}^2} \left[ \frac{T_{\mathrm{a}} - T_{\mathrm{s}}}{T_{\mathrm{a}}} + \frac{q_{\mathrm{a}} - q_s}{q_{\mathrm{a}} + \epsilon(1 - \epsilon)} \right] \tag{16}$$

$$f_z = \frac{1}{4} \sqrt{\frac{\mu}{z_{\mathrm{a}}}} \tag{17}$$

where $g$ is the gravitational acceleration (m s$^{-2}$) and $\epsilon$ is the ratio of molecular weights between water and dry air. By default, surface roughness ($\mu$) evolution is modelled after Mölg et al. (2012) as a linearly increasing function of time $t$ since the last snowfall event. However, we considered this unrepresentative of the snow surface conditions at CG that are more heavily influenced by scouring during extreme wind events. Therefore, we use a constant value of 0.001 m, derived by Suter et al. (2004) from wind profiles measurements at Seserjoch.

### 3.1.3 Precipitation Model

In order to represent the extreme spatial gradient of snow accumulation at the site, we replace the spatially homogeneous precipitation input with a three-phase anomaly method that was previously employed to the site by Mattea et al. (2021). As the dimensionality of the model cannot represent lateral mass transportation, this technique artificially adjusts nodal precipitation to accommodate for preferential deposition and losses from wind scouring to enable a representative accumulation rate. This is achieved by temporally down-scaling a spatially variable annual accumulation climatology into an hourly precipitation time series with adjustments for annual variability. Specifically, for a given grid cell $(x, y)$, at simulation time step $t$, in year $i$, precipitation is determined as

$$R(x, y, t) = C(x, y) A(i) D(t) \tag{18}$$

where $C$ is the long term annual accumulation climatology (cm w.e. yr$^{-1}$), $A$ is the annual anomaly and $D$ is the temporal downscaling coefficient. The exact formulation of the accumulation climatology raster (Fig. 3), derived from GPR profiles (Konrad, 2011) and firn cores (Lier, 2018; Licciulli et al., 2020), and the annual anomaly is explained in greater detail in



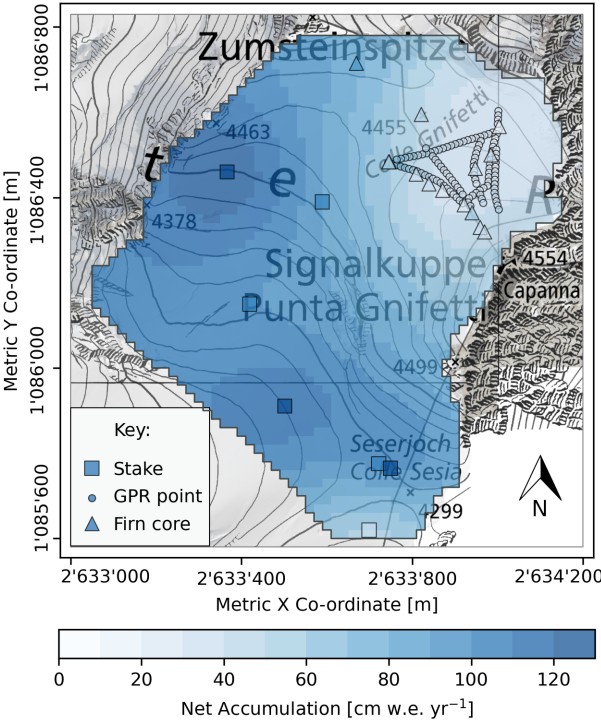

**Figure 3.** Annual accumulation climatology forming the spatial component of the precipitation model (Mattea et al., 2021). Spatial co-ordinates are defined by EPSG 2056 (Metric Swiss CH1903+/LV95). Topographic map source: Swisstopo (2017).

Mattea et al. (2021). In lieu of available hourly precipitation data directly from the AWS CM, the downscaling coefficient ($D$)
is calculated by normalising to a unit sum the average hourly precipitation rates from the closest rain gauges at Monte Moro,
Bochetta della Pisse and Rifugio Macugnaga Zamboni (Fig. 1). The model then uses a linear logistic transfer function based
on air temperature to differentiate between solid and liquid precipitation. The proportion of snowfall therefore scales between
100% at 0 °C and 0% at 2 °C (Hantel et al., 2000). By default, the density of newly added fresh snow layers ($\rho_0$) is determined
as a function of air temperature and wind speed after Vionnet et al. (2012). However we reverted to using a constant value,
given the incompatibility and complications of using this approach coupled with our artificial precipitation model.

## 3.2 Subsurface Model

Subsurface layers are discretised according to an adaptive Lagrangian re-meshing algorithm within COSIPY: layers can trans-
late vertically in the grid following mass exchange at the surface and are merged with adjacent layers if user-defined temper-
ature or density thresholds are exceeded. We found subsurface layer properties to be particularly sensitive to the selection of
the re-meshing algorithm, with the adaptive method imparting a large positive density bias against observed firn core profiles
(Appendix A). Therefore, we adopted an alternative approach: instead of creating new subsurface layers on every precipitation



**Table 3.** Model parameter values chosen for model implementation at CG compared against those used by Mattea et al. (2021) with the Energy Balance Firn Model (EBFM) of van Pelt et al. (2012). A brief explanation or reference for parameter selection is also provided.

| Parameter | Definition | COSIPY Value | EBFM Value | Unit | Source/Rationale |
|---|---|---|---|---|---|
| $a$ | Cloud transmissivity coefficient | 0.233 | 0.233 | – | Greuell et al. (1997) |
| $b$ | Cloud transmissivity coefficient | 0.415 | 0.415 | – | Greuell et al. (1997) |
| $\alpha_{snow}$ | Fresh snow albedo | 0.81 | 0.83 | – | Tuned from Seserjoch & Colle de Lys data |
| $\alpha_{firn}$ | Firn albedo | 0.55 | 0.55 | – | Mölg et al. (2012) |
| $t^*_{wet}$ | Decay timescale (melting surface) | 10 | 10 | days | Mattea et al. (2021) |
| $t^*_{dry}$ | Decay timescale (dry snow surface at 0°C) | 30 | 30 | days | Bougamont et al. (2005) |
| $K$ | Increase in $t^*_{dry}$ at negative temperatures | 5 | 14 | day °C$^{-1}$ | Tuned from Seserjoch & Colle de Lys data |
| $T_{max, t^*}$ | Temperature threshold for $t^*_{dry}$ increase | -10 | -10 | °C | Bougamont et al. (2005) |
| $c_{emission}$ | Longwave emission constant | 0.42 | 0.42 | – | Mattea et al. (2021); Suter et al. (2004) |
| $\epsilon_{clouds}$ | Cloud emissivity | 0.96 | 0.96 | – | Mattea et al. (2021); Suter et al. (2004) |
| $\mu$ | Surface roughness | 0.001 | 0.001 | m | Suter et al. (2004) |
| $z_{lim}$ | Preferential percolation depth | 3.5 | 4 | m | Tuned to CG thermistor profiles |
| $Q_{basal}$ | Basal (geothermal) heat flux | 0.035 | 0.04 | W m$^{-2}$ | Lüthi and Funk (2001) |
| $\rho_0$ | Fresh snow density | 250 | 350 | kg m$^{-3}$ | Tuned to CG firn core density profiles |

event as with the adaptive algorithm, the new approach instead collates new accumulation into the uppermost layer until a fixed height threshold ($h_{max}$) is reached and prohibits any layer merging.

The model defines subsurface layers according to their volumetric fraction of ice ($\phi_{ice}$), water ($\phi_{water}$) and air ($\phi_{air}$) (Eq. 19), with their inherent physical properties (eg. thermal conductivity ($k$)) derived as a weighted sum of their fractional composition (Eq. 20):

$$\phi_{ice} + \phi_{water} + \phi_{air} = 1 \qquad (19)$$

$$k = k_{ice}\,\phi_{ice} + k_{water}\,\phi_{water} + k_{air}\,\phi_{air} \qquad (20)$$

**3.2.1   Percolation Scheme**

COSIPY employs a 'bucket approach' percolation scheme whereby liquid water filters down into subsequent layers upon exceeding the layer saturation capacity. We supplemented this with the statistical deep preferential percolation scheme of Marchenko et al. (2017) due to our assessment that the default bucket approach underestimates the true percolation depth in a meltwater sparse environment. The Gaussian method was selected, as used in the EBFM, that instantly distributes all




surface water in accordance with a normal Probability Density Function (PDF) up to a pre-defined characteristic preferential percolation depth ($z_{\mathrm{lim}}$):

$$
\mathrm{PDF}_{\mathrm{normal}}(z, z_{\mathrm{lim}}) = 2 \left[ \frac{\exp\left(-\frac{z^2}{2\sigma^2}\right)}{\sigma\sqrt{2\pi}} \right]
\tag{21}
$$

$$
\sigma = z_{\mathrm{lim}} / 3
\tag{22}
$$

where $\sigma$ is the standard deviation of the probability density function and $z_{\mathrm{lim}}$ represents the pre-defined characteristic preferential percolation depth. We tuned this value to 3.5 m to match 20 m depth firn temperatures at CG.

Penetrating shortwave radiation (see section 3.1.1) can also directly melt the ice matrix of subsurface layers, if they are warmed to the melting temperature, supplementing their water content. Subsurface water can subsequently refreeze if there is sufficient cold content and volumetric capacity in layers – the latter being limited by the set pore closure density of 830 kg m$^{-3}$ at the transition from firn to glacial ice. Any remaining water is stored by capillary and adhesive forces (irreducible water content) with the excess being drained into deeper layers via the bucket scheme. The maximum irreducible water ($\theta_{\mathrm{irr}}$) is based on the volumetric ice fraction of subsurface layers (Coléou and Lesaffre, 1998; Wever et al., 2014):

$$
\theta_{\mathrm{irr}} = \begin{cases}
9.0264 + 0.0099\frac{(1-\phi_{\mathrm{ice}})}{\phi_{\mathrm{ice}}}, & \phi_{\mathrm{ice}} \leq 0.23 \\
0.08 - 0.1023(\phi_{\mathrm{ice}} - 0.03), & 0.23 > \phi_{\mathrm{ice}} \leq 0.812 \\
0, & \phi_{\mathrm{ice}} > 0.812
\end{cases}
$$

Water reaching the base of the computational domain is instantly removed and recorded as run-off.

### 3.2.2 Thermal Diffusion

The evolution of the subsurface thermal regime is driven by the processes of thermal diffusion, water refreezing and subsurface melting:

$$
\rho\, c_{\mathrm{p}} \frac{\delta T}{\delta t} = \frac{\delta}{dz}\left(k\frac{\delta T}{dz}\right) + L_{\mathrm{m}}(F - M)
\tag{23}
$$

where $\rho$, $c_{\mathrm{p}}$, $k$ are the effective density (kg m$^{-3}$), specific heat capacity (J kg$^{-1}$ K$^{-1}$) and thermal conductivity (W m$^{-1}$ K$^{-1}$) of subsurface firn layers, $L_{\mathrm{m}}$ is the latent heat of melting (J kg$^{-1}$) and $F$ and $M$ are the refreezing and subsurface melting rates (kg m$^{-3}$ s$^{-1}$) respectively. By default, COSIPY resolves equation (23) with a fixed temperature at the base of the modelling domain, however we substituted this for a basal (geothermal) heat flux ($Q_{\mathrm{basal}}$). The effective thermal properties of subsurface firn layers are derived in the baseline model according to the bulk-volumetric approach (Eq. 20). However, this



was found to significantly overestimate thermal conductivity, particularly at low densities, leading to an inability for the firn to
retain thermal energy. Therefore we converted to using the empirical formulation of Calonne et al. (2019) that defines it as:

$$k(\rho, T) = (1 - \vartheta) \frac{k_i(T) k_a(T)}{k_i^{\mathrm{ref}} k_a^{\mathrm{ref}}} k_{\mathrm{snow}}^{\mathrm{ref}}(\rho) + \vartheta \frac{k_i(T)}{k_i^{\mathrm{ref}}} k_{\mathrm{firn}}^{\mathrm{ref}}(\rho) \tag{24}$$

$$\vartheta = 1 / \left[ 1 + \exp(-2a(\rho - \rho_{\mathrm{transition}})) \right] \tag{25}$$

$$k_{\mathrm{firn}}^{\mathrm{ref}} = 2.107 + 0.003618(\rho - \rho_i) \tag{26}$$

$$k_{\mathrm{snow}}^{\mathrm{ref}} = 0.024 - 1.23\rho \times 10^{-4} + 2.5 \times 10^{-6} \rho^2 \tag{27}$$

where $k_i(T)$ and $k_a(T)$ are the ice and air thermal conductivity at the temperature T, $k_i^{\mathrm{ref}}$ = 2.107 W m$^{-1}$ K$^{-1}$ and $k_a^{\mathrm{ref}}$
= 0.024 W m$^{-1}$ K$^{-1}$ are the ice and air thermal conductivities at the reference temperature of -3 °C, $a$ = 0.02 m$^3$ kg$^{-1}$ and
$\rho_{\mathrm{transition}}$ = 450 kg m$^{-3}$. We also utilised the empirical, temperature-dependant specific heat capacity of Yen (1981):

$$c_{\mathrm{p}}(T) = 152.2 + 7.122\,T \tag{28}$$

### 3.2.3   Firn Densification

By default, dry firn densification is modelled in COSIPY after Boone (2009), accounting for compaction from overburden
pressure and the thermal metamorphosis of snow. As a result of difficulties encountered using this method to generate firn
density profiles representative of those observed at CG, we switched to the semi-empirical method of Arthern et al. (2010)
based on the processes of sintering and lattice-diffusion creep of consolidated ice. An enhancement by Ligtenberg et al. (2011),
furthers this by adding a dependence on the local accumulation rate:

$$\frac{d\rho}{dt} = C\,c_{\mathrm{lig}}\,g(\rho - \rho_{\mathrm{ice}}) \exp\left(-\frac{E_c}{RT} + \frac{E_g}{R\overline{T}}\right) \tag{29}$$

$$c_{\mathrm{lig}}(C, \rho) = \begin{cases} 0.0991 - 0.0103 \log(C), & \rho < 550\,\mathrm{kg\,m^{-3}} \\ 0.0701 - 0.0086 \log(C), & \rho \geq 550\,\mathrm{kg\,m^{-3}} \end{cases} \tag{30}$$

where $C$ is the accumulation rate (mm yr$^{-1}$), $\rho$ is the layer density (kg m$^{-3}$), $T$ is the current layer temperature (°C), $\overline{T}$ is
the average layer temperature of the preceding year (°C), $R$ is the universal gas constant (J mol$^{-1}$ K$^{-1}$) and $E_c$ = 60 kJ mol$^{-1}$
and $E_g$ = 42.4 kJ mol$^{-1}$ are the activation energies associated with creep by lattice diffusion and grain growth respectively. The
processes of meltwater refreezing and subsurface melting also change the fractional constituents of subsurface layers, thereby
altering their density into addition to the process of dry firn densification.

### 3.3   Model Initialisation

In order to initialise the model subsurface to steady state conditions, the main simulation was preceded by a 64 year spin-up
using eight loops of a reconstructed 1995-2002 meteorological forcing. Representative months were selected from the existing





meteorological time series based on their correlation to the reconstructed Mean Monthly Air Temperature (MMAT) at CM, calculated using quantile mapping of the air temperature series from Jungfraujoch (3,454 m a.s.l.) (MeteoSwiss, 2024). The simulation depth was set to a limit of 30 m using subsurface layers up to a maximum height of 10 cm. Beyond a threshold depth of 21 m, coarser 1 metre height layers are used for greater computational efficiency, giving an average of 300 layers per node due to layer compaction. After initialisation, the main simulation runs with an hourly temporal and 20 m spatial resolution for a 21 year period between 2003 and 2024.

## 4  Results

While COSIPY outputs a large selection of surface and subsurface variables, in this section we predominantly focus on those that influence the thermal regime of a cold firn facie. Starting with the simulated surface energy exchanges and the resultant quantity of surface melt produced, we proceed to report on the energy release from deep meltwater percolation and the subsequent diffusion of heat to depth that leads to sustained firn warming.

### 4.1  Surface Energy & Mass Exchanges

There is strong spatial variability in modelled surface exchanges across CG (Fig. 4), driven by the topographically-derived insolation gradient. The exposed, south-facing aspect of the Zumsteinspitze slope (ZS) receives the greatest shortwave flux and correspondingly has to compensate with a greater expenditure of energy through melt compared to the Signalkuppe slope (SK). However, melt is mostly limited to the summer months at CG and even in July the melt flux does not exceed 6% of the monthly energy turnover at the saddle point (SP). Turbulent exchange also makes a significant contribution to monthly energy turnover at CG, although it has a high temporal variability due to the dependence on wind speed in the model parameterisation (Eqs. 11-12). During winter at SK, sensible heat is an even greater energy source than shortwave radiation due to the steep temperature gradient modelled between the surface and the air. Sensible exchange is conversely less influential at ZS in winter as the surface temperature is higher due to greater insolation. The prevalence of cold, dry air in the meteorological forcing means the endothermic process of sublimation represents 87% of turbulent mass exchange. Deposition, by contrast, accounts for the remaining 13% and typically requires the relative humidity to exceed 90% to reverse the prevailing moisture gradient. Therefore, net latent exchange acts predominantly as an energy sink and plays a vital role in inhibiting surface melt, given that the magnitude of longwave emission is physically limited by the surface temperature constraints. Indeed, the simulation of surface melt occurs almost exclusively at wind speeds below 5 m s$^{-1}$, when latent exchange is limited. However, despite the much greater dissipation of energy via latent exchange, annual surface mass ablation via sublimation rarely exceeds melt as a result of the latent heat of sublimation ($2.83 \times 10^6$ J kg$^{-1}$) being an order of magnitude greater than melting ($3.34 \times 10^5$ J kg$^{-1}$).



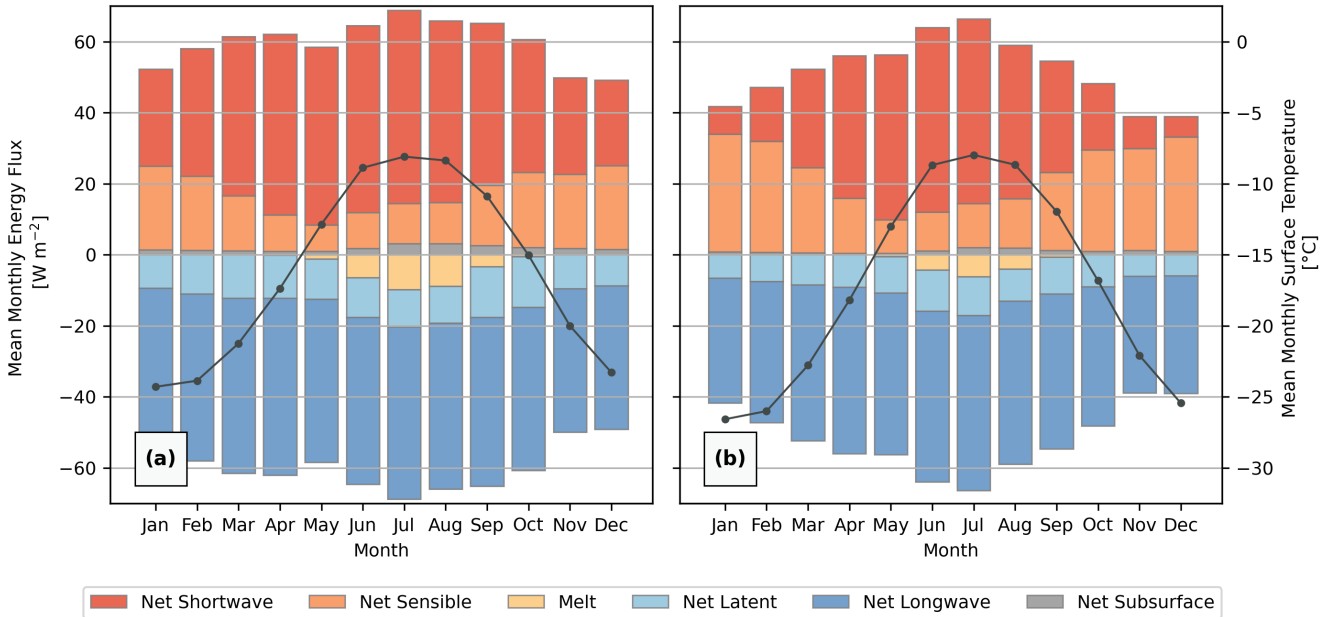

**Figure 4.** Modelled mean monthly surface energy fluxes (2003-23) at the **(a)** Zumsteinspitze Slope and the **(b)** Signalkuppe Slope with mean monthly surface temperature overlain on the secondary y-axis (20 m spatial & 1 hour temporal resolution). The SEB of the saddle point is not displayed but possesses approximately intermediary values between the displayed locations.

## 4.2 Melt

The spatial distribution of modelled mean annual surface melt (Fig 5a) largely corresponds with the patterns of modelled surface exchanges, with variation strongly influenced by differences in nodal elevation, aspect and slope. Across the reference locations at CG, it ranges between 11.6 cm w.e. $yr^{-1}$ at the SK, 16.1 cm w.e. $yr^{-1}$ at the SP and 24.0 cm w.e. $yr^{-1}$ at the ZS, with the lower altitude slopes of the Grenzgletscher reaching a maximum of 37.4 cm w.e. $yr^{-1}$. Conversely, some of the north-facing margins of the Signalkuppe have negligible to no modelled surface melt, indicative of the re-crystallisation firn

facie zone. Despite strong inter-annual fluctuations, statistically significant trends of increasing annual surface melt can be identified at CG - particularly if the extreme melt year of 2003 is not considered (Fig 5b). Linear least squares regression from 2004-2023 yields an increase in surface melt ranging from 0.43 cm w.e. $yr^{-2}$ at the SK, 0.54 cm w.e. $yr^{-2}$ at the SP and 0.74 cm w.e. $yr^{-2}$ at the ZS. Annual variation in modelled melt holds a strong Pearson's correlation of 0.89 to the Mean Annual Summer Air Temperature (MASAT) recorded at the CM AWS, which increases by 0.10 °C $yr^{-1}$ over this time period.

Modelled subsurface melt generally adheres to the same spatio-temporal patterns of surface melt, but accounts for an average of only 15.1% of combined surface and subsurface melt at the SP. The process is mostly confined to the uppermost metre of the firn, with 82.3% of simulated subsurface melt events happening on time-steps with surface melt occurring simultaneously. The remainder predominantly occur following surface melt events in clear sky conditions, when abrupt increases in wind speed





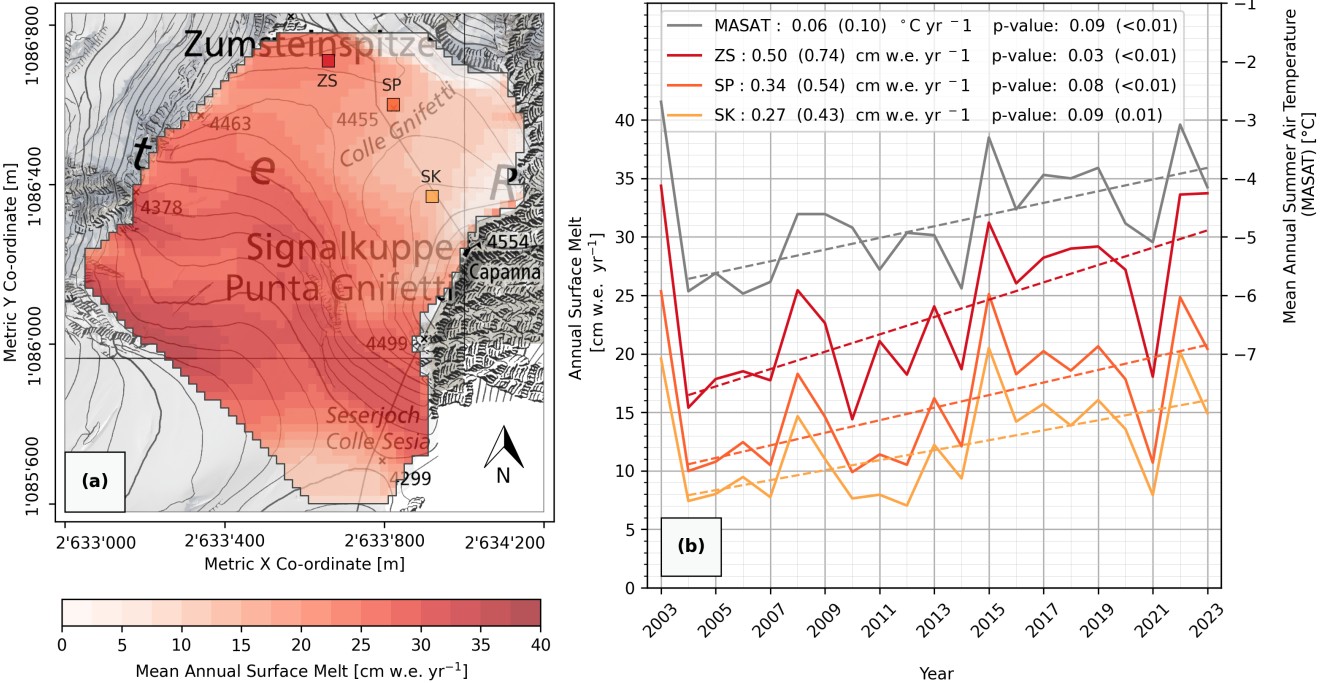

**Figure 5. (a)** Modelled mean annual surface melt (2003-2022). **(b)** Modelled annual surface melt (2003-2022) at the three reference locations (solid lines; ZS : Zumsteinspitze Slope, SP: Saddle Point, SK: Signalkuppe Slope) compared against Mean Annual Summer Air Temperature (MASAT). Linear regression trends, calculated excluding the extreme melt year of 2003, are displayed as dashed lines. Annual trends are reported for the full 2003-23 simulation period (first number) and excluding the extreme melt year of 2023 (second number, in parenthesis). Spatial co-ordinates are defined by EPSG 2056 (Metric Swiss CH1903+/LV95). Topographic map source: Swisstopo (2017)

suppresses simulated melt at the surface. Notably, the simulation of surface melt is not restricted to time-steps with positive air temperatures; 19.4% of the total melt recorded at the SP occurs during sub-zero air temperatures.

## 4.3 Firn Temperatures

Whilst simulated near-surface firn temperatures are subject to a high degree of inter-annual fluctuation due to their strong coupling to the applied meteorological forcing, progressively deeper layers become more thermally isolated (Fig 6). Simulated meltwater-refreezing events can be visually identified in Fig. 6 by steep protrusions of near-zero degree temperatures into the depths of the firn following summer melt events - particularly evident at the ZS in Fig. 6a. In contrast, the magnitude of these events is considerably smaller at the SP and SK, resulting in reduced firn warming. The depth of meltwater refreezing events is constrained in the simulation by the percolation parameterisation of Marchenko et al. (2017) and the characteristic depth $z_{lim}$ set to 3.5 metres, but exceedance of this limit is possible if all overlying layers become saturated. Nevertheless, this is marginal at these reference locations and 89.2% of simulated refreezing occurs above a depth of 3.5 metres at CG.





**Figure 6.** Modelled firn temperature-depth profiles at the **(a)** ZS: Zumsteinspitze slope, **(b)** SP: Saddle Point and the **(c)** SK: Signalkuppe Slope (2003-2022) (20 m spatial & 1 hour temporal resolution). Zero Annual Amplitude (ZAA) ($\Delta T < 0.1$ °C) depth is overlain as a line graph. Firn warming rates over the 21 year simulation at 20 metre depth ($\Delta T_{20m}$) are also indicated on the figure.

Thermal diffusion subsequently transports thermal energy to the full depth of the simulated domain following a lag period of approximately 2 years. At a depth of 20 metres, modelled annual firn temperature variation rarely exceeds ± 0.1 °C, except following extreme melt years such as 2003, 2015 and 2022 and is therefore considered indicative of long-term climatic changes.



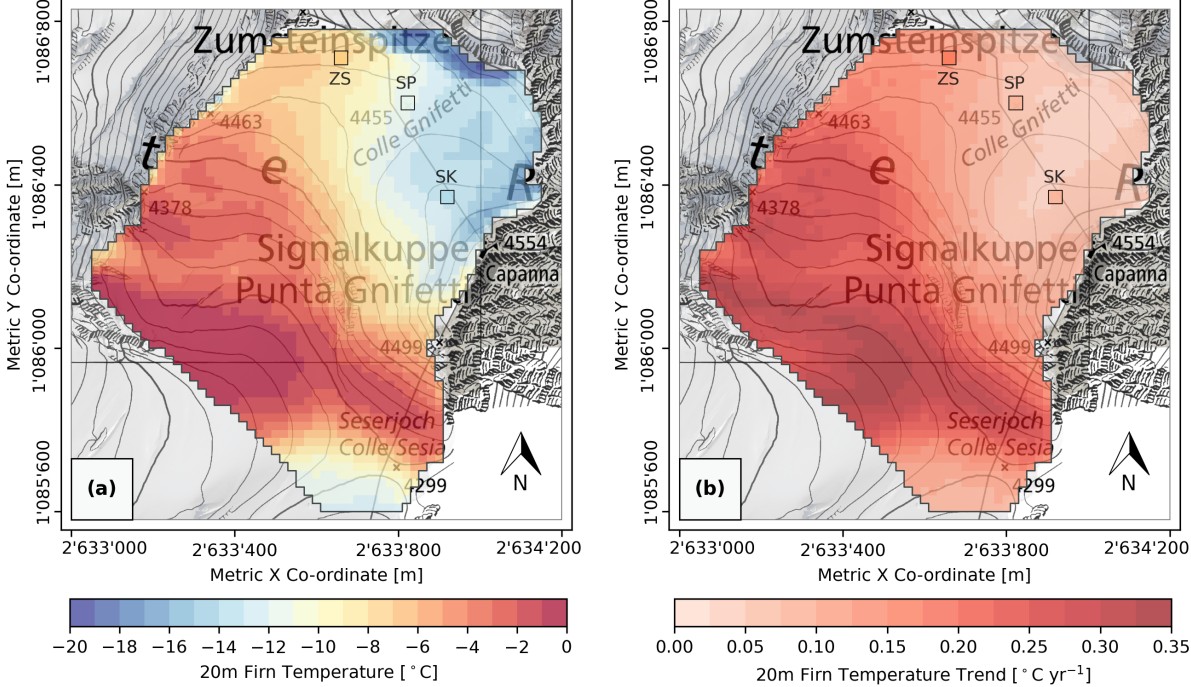

**Figure 7. (a)** Modelled 20 m depth firn temperatures at the simulation end date of the 31$^{\text{st}}$ December 2023 and **(b)** 20 m depth firn evolution during the full 21 year simulation range (2003-2023) (20 m spatial & 1 hour temporal resolution). Spatial co-ordinates are defined by EPSG 2056 (Metric Swiss CH1903+/LV95). Topographic map source: Swisstopo (2017)

Corresponding with the spatial patterns of surface melt, there are similarly strong gradients in the modelled 20 m depth firn
temperature that generally reflect changes in elevation and surface aspect (Fig 7a). At CG, temperatures range between -13.20
°C at the SK and -6.54 °C at the ZS over a lateral distance of less than 500 m – a temperature gradient of 1.54 °C 100 m$^{-1}$ .
On the lower altitude, south-facing slopes of the Grenzgletscher, towards the south-western region of the spatial domain, mod-
elled deep firn temperatures reach that of a temperate firn facie by the summer of 2022 – coinciding with the area of greatest
simulated surface melt (Fig 5a).


Simulated firn warming at the SP is 0.056 °C yr$^{-1}$ equating to a 1.18 °C increase over the course of the 21 year simulation.
There is considerable variation across the saddle consistent with the previously displayed spatial patterns of melt and temper-
ature, however these changes are modest compared to the lower elevation regions (Fig 7b). In these areas at the south-western
reaches of the domain, englacial warming rates are much greater in excess of 0.3 °C yr$^{-1}$, four times the local recorded rate
of atmospheric warming of 0.073 °C yr$^{-1}$ from the CM AWS time series – calculated using linear least squares regression of
MAAT (p value < 0.01). The irregularities in the warming rates in this area are as of result of some locations reaching temperate
conditions (Fig 7a), where further warming is physically constrained as it attains the melting point.



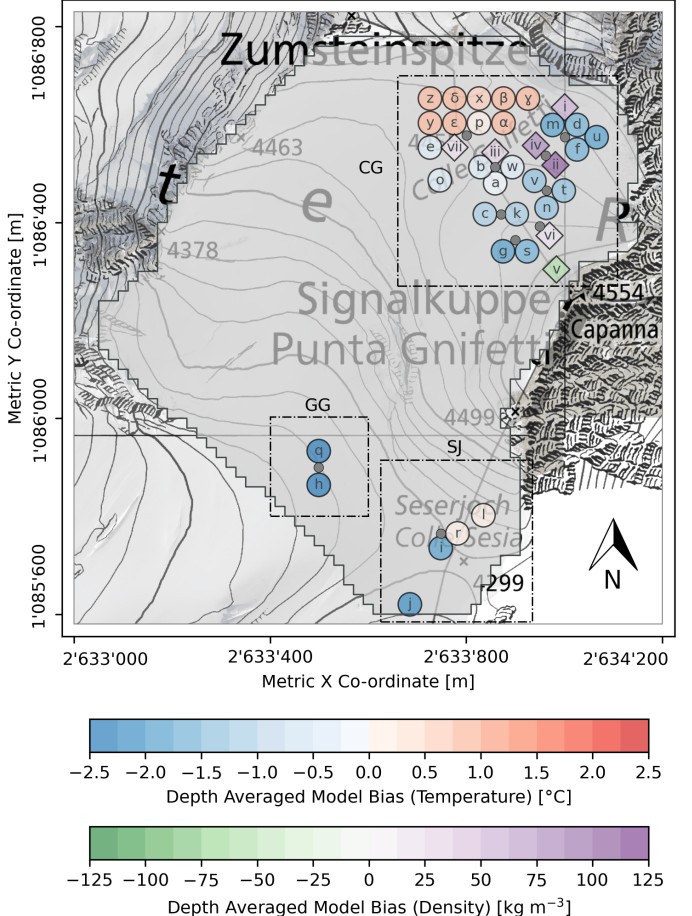

**Figure 8.** Depth-averaged model temperature and density bias across the computational domain. Borehole thermistor profile codes have the following designation: LLYY-I/yy, where: 'LL' indicates the borehole location - either Colle Gnifetti [CG], Seserjoch [SJ] or Grenzgletscher [GG]; 'YY-I' gives the drilling year and index number of the borehole ; and 'yy' specifies the year the temperature profile was measured (Hoelzle et al., 2011; GLAMOS, 2020, 2022). Spatial co-ordinates are defined by EPSG 2056 (Metric Swiss CH1903+/LV95). Topographic map source: Swisstopo (2017)

## 4.4 Model Validation

Depth-averaged model firn temperature bias ranges between -2.35 °C and 0.91 °C, when compared to borehole thermistor
measurements (Fig. 8). This corresponds to an average of -0.65 °C or a Root Mean Square Error (RMSE) of 1.28 °C. Spatially, models residuals show a tendency to significantly underestimate firn temperatures around the periphery of the Signalkuppe flank at CG and at lower elevations at the Seserjoch (SJ) and Grenzgletscher (GG). In the case of the latter two locations, the model initialises firn temperatures several degrees lower than historic measurements from 1999 (Suter and Hoelzle, 2002) –



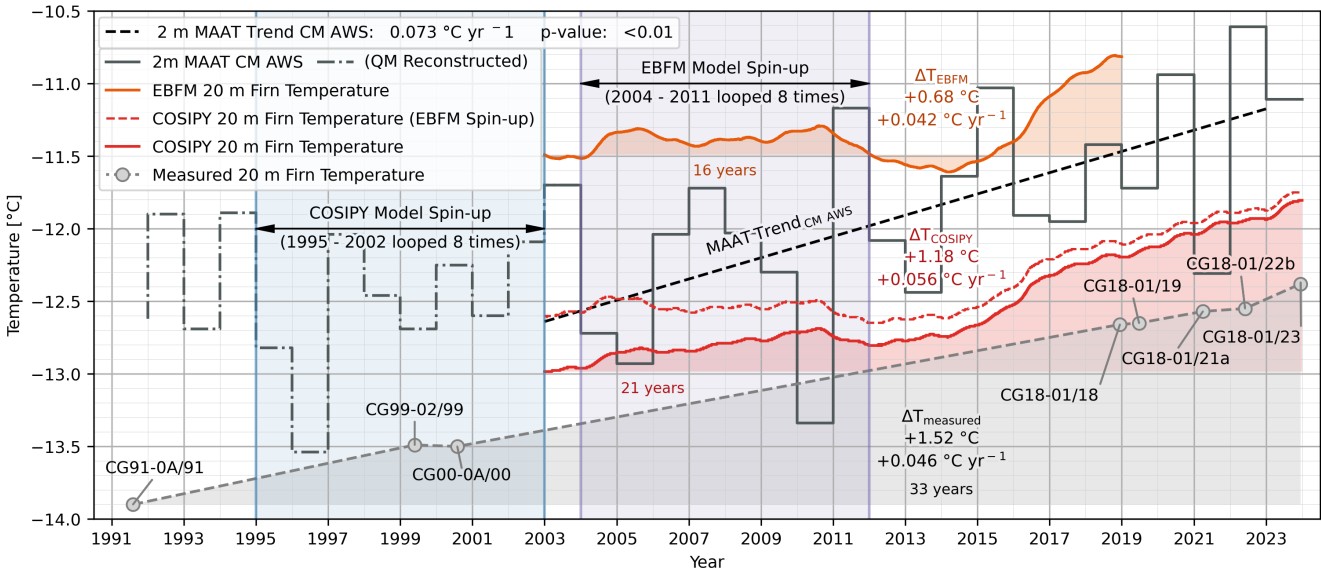

**Figure 9.** Measured englacial warming at the CG SP compared to the COSIPY simulation (Haeberli and Funk, 1991; Suter et al., 2001; Suter and Hoelzle, 2002; GLAMOS, 2020, 2022), previous modelling efforts using the EBFM of van Pelt et al. (2012) (Mattea et al., 2021) and the Mean Annual Air Temperature (MAAT) from the CM AWS meteorological time series. Meteorological data prior to the establishment of the CM AWS in 2003 is reconstructed using quantile mapping of the Jungfraujoch (3,454 m a.s.l.) air temperature series (MeteoSwiss, 2024). Borehole thermistor profile codes adhere to the common naming convention described in the caption of Fig. 8.

preceding the simulation temporal range – which is counterbalanced by an excessive warming rate (Fig. 7b). Simulated density
profiles on average have a slight positive density bias of 39.1 kg m$^{-3}$ and RMSE of 93.9 kg m$^{-3}$, however firn core profiles are spatially limited to a small extent of the modelling domain around CG.

## 4.5 Englacial Warming at Colle Gnifetti

The simulation results can also be assessed temporally against historic englacial temperature measurements from CG. Since
1991, measured 20 m depth temperatures at the SP have increased by 1.52 °C at a near-linear rate of 0.046 °C yr$^{-1}$ over the 33 year period (Figs. 9 & 10); the previous decade since the 1982 measurement ('CG82-01/81') having less than a 0.1 °C change. Notably, the rate of observed warming is considerably slower than the 21 year trend in Mean Annual Air Temperature (MAAT) at the CM AWS from 2003 of 0.073 °C yr$^{-1}$ (p value < 0.01). However, the QM reconstruction of the air temperature series for CM using data from Jungfraujoch (3,454 m a.s.l.) indicates there is no evidence of a significant trend from 1991-2002.

Simulated firn warming by COSIPY is 21% greater in magnitude (0.056 °C yr$^{-1}$) with an average temperature bias of +0.4 °C. Figure 9 also demonstrates that the selection of the model spin up can greatly influence the nature of modelled firn warm-



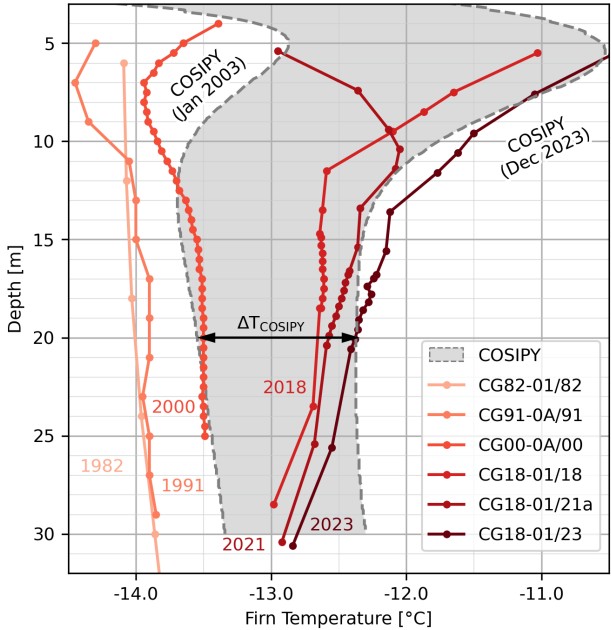

**Figure 10.** Bias-corrected simulated englacial warming (grey enclosed region) compared against observed borehole thermistor profiles extracted from the CG SP (Haeberli and Funk, 1991; Suter et al., 2001; Suter and Hoelzle, 2002; GLAMOS, 2020, 2022). Borehole thermistor profile codes adhere to the common naming convention described in the caption of Fig. 8.

ing, as using meteorological data from 2004-2011 (as employed by the previous research of Mattea et al. (2021)) initialises firn temperatures 0.38 °C higher. This results in limited warming in the initial 11 years of the simulation as opposed to the prolonged trend displayed using our spin-up with meteorological data based on MMAT from 1995-2002.

A marked rotation in the thermal gradients of measured profiles beyond the depth of seasonal variation ($\sim$ 15 m) is observed in Fig. 10. Between 1982 and 2000, observed firn temperature in the 15 - 30 m depth range were near constant, however by 2023 a distinct negative gradient of -0.042 °C m$^{-1}$ is visible from the 'CG18-01/23' profile. COSIPY also shows an alteration to the thermal gradient in the uppermost 30 m of the firn, but the simulation starts with a steeper gradient representative of steady state conditions from the model spin-up. At the SP, only the 'CG82-01/82' profile extends beyond 30 m to the glacier base at a depth of approx. 120 m, where the basal firn temperature tends to -12.3 °C (Haeberli and Funk, 1991). Evidence from more recent deeper thermistor profiles taken towards the Signalkuppe flank ('CG21-01/21') suggest warming is currently less pronounced beyond 40 m and following an inflexion point in the thermal gradient, englacial temperatures continue to increase with depth towards the bedrock (Schwerzmann, 2006; GLAMOS, 2022).



## 5 Discussion

### 5.1 Melt

The COSIPY simulation shows a prolongation of the trends of increasing surface melt at CG identified by Mattea et al. (2021) and a strong corroboration with the spatial distribution of melt shown by their simulation using the EBFM (Fig 5). Nevertheless,

our results are not considered fully validatory given the operational similarity between the two applied models - particularly with regard to surface interactions. Explicit measurements of melt are deficient at CG, however Lier (2018) compiled quantitative estimations of the magnitude of refreezing by assessing observed density anomalies from firn cores at CG compared to an ideal dry densification profile. Annual refreezing rates within confidence intervals of 3 - 33 cm w.e. $yr^{-1}$ and 1 - 13 cm w.e. $yr^{-1}$ were reported for the Zumsteinkern and Sattelkern, situated at the ZS and SP reference points respectively. A wide variety

of cores were analysed on the Signalkuppe flank with values ranging between 0 - 16 cm w.e. $yr^{-1}$, however none of these are co-located with our SK reference location. These results provide reasonable attestation to our simulated melt quantities (Fig. 5a), given the inherent uncertainties in this technique and the tendency for underestimation as a result of the occurrence of melt-refreeze cycles (Mattea et al., 2021). Furthermore, given that the surface model was calibrated and produces a SEB in accordance with the energy fluxes measured at Seserjoch (Fig. 4) (Suter et al., 2004), the magnitude of the simulated melt flux

and resulting surface melt are considered reliable. Finally, the occurrence of surface melt events during negative air temperatures (also simulated by Mattea et al. (2021)), is a phenomenon that has previously been observed at this temporary weather station (Suter et al., 2004).

A recognised limitation in the architecture of COSIPY's skin layer SEB formulation is the decoupling of energy absorption

at the surface and the subsequent dissipation of thermal energy via diffusion (Brun et al., 2022). Operating at our hourly temporal frequency, the delay in the evacuation of energy from the surface leads to a potential susceptibility for significant temperature and thus melt overestimates. Brun et al. (2022) reported a 30% reduction in melt by switching to a minute timestep for the simulation of an ablating ice surface and also a high sensitivity to the selection of the subsurface heat flux parameters ($z_{\text{interp 1 \& 2}}$). However, investigations using our model with a temporally downscaled version of our meteorological time series

at 1 minute intervals produced less than a 7% variation in melt at the SP. Site differences are suspected to be a principal factor in this discrepancy as Brun et al. (2022) concluded this problem was most pronounced on the modelling of the ablation of an exposed ice surface. The uninterrupted coverage of low density snow layers with a weak thermal conductivity throughout the ablation season at CG greatly curtails the contribution of the modelled subsurface heat flux to the SEB at CG (Fig. 4) and therefore the rate of thermal diffusion at the surface is low.

### 5.2 Firn Temperatures


Modelled firn temperatures show good corroboration with in-situ measurements and the previous modelling efforts of Mattea et al. (2021), showcasing the strong aspect and elevation dependant gradients at CG previously established by Suter and Hoelzle (2002). However, the exact temperatures have a particularly high-sensitivity to the selection and parameter calibration of





the preferential percolation scheme of Marchenko et al. (2017) due to its strong influence on the retainment of thermal energy
released by meltwater refreezing. Our high resolution thermistor chains at the SP provide insight into the importance of this
process, showing a remarkable increase of 7.74 °C at 4 m depth over the course of 20 days during the summer melt season
of 2022. Indeed, failing to represent preferential percolation by using the default bucket approach of COSIPY was found to
result in an underestimation of modelled firn temperatures by as much as 10 °C on the lower Grenzgletscher slope (Appendix
A). However, the statistical approach used still has considerable limitations: principal among which being the spatially and
temporally constant characteristic preferential percolation depth ($z_{lim}$). In actuality, the percolation depth has been observed
to increase over the course of a melt season and is heavily influenced by the supply of meltwater (Marchenko et al., 2017)
and historic infiltration and refreezing events which affect the firn's stratigraphy (Illangasekare et al., 1990). Physically-based
preferential percolation schemes are generally derived by resolving the Richards equation or the simplified Darcy's law but are
typically computationally expensive (Short et al., 1995; Vandecrux et al., 2020). An estimation of grain size is also required to
derive the preferential flow channel area (Katsushima et al., 2013) but in-situ quantitative data on snow and firn micro structure
is scarce. Wever et al. (2014) implemented a dual-domain approach into the SNOWPACK model (Bartelt and Lehning, 2002),
whereby the pore space can be subdivided into both matrix and preferential flow, however its implementation into COSIPY
would require a complete redesign of the sub-surface model. Nor is there any guarantee that more sophisticated physically-
based methods would necessarily yield more accurate results; Verjans et al. (2019) noted that the performance of the Wever
et al. (2014) method did not always supplant a single domain Richard's equation scheme or a standard bucket approach.

The dimensionality of COSIPY is a further constraint for the accurate representation of percolation in general, as flow is
rarely confined solely to the vertical direction; in areas of steep inclination, gravitational forces will induce lateral movement
down-slope. Furthermore, the formation of ice lenses from historic meltwater-refreezing events can lead to the development of
low-permeability barriers to the downward percolation of water, forcing lateral flow as a consequence. The firn ultimately de-
velops a complex and highly irregular stratigraphy over time, with multiple impedances to meltwater flow forming (Machguth
et al., 2016; Clerx et al., 2022). Thus, truly accurate modelling of percolation would require a fully three-dimensional spatial
model.

Spatially, model output sensitivity is significantly greater in the south-western reaches of the computational domain, particu-
larly with regard to firn temperatures. These lower-altitude, heavily insolated areas are subject to greater meltwater production
(Fig. 5a) and as such domain-wide adjustments to the model have considerably larger implications on the magnitude and re-
tainment of latent heat release from refreezing - ultimately manifesting in greater firn temperature variance. The spatial extent
of the cold-temperate firn transition displayed in Fig. 7a is therefore subject to a large-degree of uncertainty. The tuning of
key model parameters based on conditions at the CG saddle and the dense concentration of validation data inevitably causes
a systematic bias away from this area, further exacerbating this issue. Accurately demarcating firn facie boundaries and their
temporal evolution would therefore require a computational domain encompassing a more complete range of topographic ex-
positions across altitudes between 3,800 and 4,600 m a.s.l. and a better distributed validation set – akin to the research of Suter



and Hoelzle (2002).


Variation between observed and simulated firn temperatures could also be attributed to physical processes not represented in the COSIPY model. The eastern flank of the saddle and the Signalkuppe comprises of a strongly insolated vertical rockwall, measured by Gruber et al. (2004) to have a Mean Annual Ground Surface Temperature (MAGST) of -5.4 °C in 2001. Assuming basic one-dimensional heat conduction through ice, the four profiles of the 'CG05-01' borehole that are located

150 m away could be subject to a lateral heat flux of 0.1 W m$^{-2}$ – significantly greater than the prescribed vertical basal heat flux of 0.035 W m$^{-2}$ and potentially influential in the model's underestimation of temperatures on the periphery of the Signalkuppe slope (Fig. 8). Further three-dimensional effects such as heat advection are also not considered. Whilst measured surface velocities around the SP are small at a magnitude of 1-3 m yr$^{-1}$, these increase on the southeastern reaches of the saddle with an acceleration of ice flow moving down onto the steeper slopes of the lower Grenzgletscher (Licciulli et al., 2020).


With regard to englacial warming at CG, while this is undoubtedly strongly associated with the observed trend in increasing MAAT (Fig 9), there are significant limitations using this metric to assess firn changes. Given that melt occurs almost exclusively during the summer at CG and the profound influence it has on the thermal regime, the summer air temperature is fundamentally much more indicative of deep firn temperature evolution than the annual mean. In our simulation, the MASAT

has been demonstrated to strongly correlate with annual melt and subsequent firn warming (Figs. 5b & 6) in comparison to the MAAT (eg. 2007, 2011, 2021). Whilst this can be partly attributed to the simplistic approach of using an annual mean, it also a reflection of the stronger influence of the summer melt season. The meltwater infiltration re-crystallisation process enables energy to bypass the thermally insulating, near-surface firn layers that effectively decouple deep firn temperatures from extreme conditions at the surface. As such firn temperatures at CG are much more sensitive to small variations in summer melt,

than any extreme meteorological conditions found throughout the remainder of the year. This is the principal reason why our model-spin up was reconstructed based on a reconstruction of monthly air temperatures, to ensure our meteorological data was representative of seasonal conditions prior to 2003.

It is evident from our investigation that small alterations in the meteorological data used in the model spin-up have a strong

influence on the magnitude of modelled firn warming (Fig. 9); sometimes of greater significance than major changes to the parameterisations of the model's physical processes. Initialising the model to accurate steady-state conditions is an important pre-requisite to prevent a transient response during the initial time period of any simulation; in our case, preliminary simulation runs using 2004-2011 meteorological data for the spin-up heavily influenced initial firn temperature evolution. In actuality, the subsurface thermal regime at CG has not been in equilibrium since the 1980s and is continually evolving in response to

atmospheric warming at the surface over the previous four decades (Haeberli and Funk, 1991; Lüthi and Funk, 2001). Thus, an accurate representation of this in our model would require an extended meteorological forcing of a further 20 years and potentially an extended simulation depth. This is evidenced by the disparity between the observed and simulated thermal gradients in Fig. 10, whereby the 'CG00-0A/00' profile has clearly progressed beyond steady state conditions in contrast to the



initial model state. Nonetheless, it is therefore our recommendation that great consideration should be taken to the accuracy of
the model spin-up and the resulting initial conditions, particularly when modelling changes to the thermal regime of firn.

### 5.3    Future Firn Evolution at Colle Gnifetti

The observation of a near-linear trend of englacial warming over the course of 30 years may suggest a prolongation of currently
observed trends when predicting the evolution of firn temperatures at CG. However, our simulation results and observations
from the lower altitude Seserjoch (4,299 m a.s.l.) may be a better indicator of future conditions at the higher altitude CG, with
firn warming accelerating in line with rising air temperatures. Although, at the Col du Dôme site of the Mont Blanc massif
(4,250 m a.s.l.), observations have shown a more non-linear response of englacial temperature to atmospheric warming in the
last three decades (Vincent et al., 2020). Following a warming of 2 °C between 1994 and 2010, a significant cooling in near-
surface temperatures has been observed. This was interpreted by Vincent et al. (2020) to be a consequence of the formation of
low-permeability ice layers that limit the percolation depth of meltwater and consequently the retainment of latent heat release
from refreezing. With our trend of increasing surface melt and evidence of small ice lenses already being observed by Mattea
et al. (2021) in 2019, the establishment of such a negative feedback affecting future englacial temperature change at CG is also
a distinct possibility. It is also evident that this further reinforces the importance of employing a more sophisticated percolation
scheme in order to accurately simulate the evolving thermal regime in a cold firn region.

## 6    Conclusions

Our research shows that there is considerable potential for using the COSIPY model to accurately model the evolution of cold
firn in complex mountainous topography. However, this could only be achieved following significant modifications to several
of the model's parameterisations for key physical processes. It is as of yet unclear if the difficulties encountered are site-specific
or relevant to applying the model to cold firn in general.

The adapted model results show good agreement with the spatial patterns of observed firn temperatures around CG and a
prolongation of a previously identified trend of increasing surface melt (Mattea et al., 2021). This influx of additional meltwa-
ter and the resulting latent heat release from refreezing at depth leads to the simulation of firn warming at a rate of 0.056 °C
yr$^{-1}$ at the saddle point, which is comparable to in-situ measurements. While observed englacial warming trends at CG have
evolved near-linearly since 1991, the simulation results from lower altitude areas provide insight into possible future conditions
at the saddle. Around the Seserjoch and lower Grenzgletscher ($\sim$ 4,300 m a.s.l.), simulated warming is much greater than the
local rate of atmospheric warming resulting in a rapid transition from cold to temperate firn. However, owing to the scarcity of
validation data and the model's sensitivity, there is high uncertainty in the results for this area.

The study mainly reveals the challenges of accurately modelling firn facie transition, as small changes in the parameterisa-
tions of the sub-surface model can lead to large temperature variance. In particular, accurately quantifying the refreezing depth



of infiltrating meltwater is pivotal in controlling the retainment of thermal energy within the firn. While a variety of viable percolation schemes exist, their performance can be highly variable and their implementation can be constrained by a model's architecture and access to site-specific calibration data. The setup and meteorological data used for the model spin-up is also found to be of profound importance; particularly when analysing the evolution of the sub-surface thermal regime in firn.

## Appendix A:  Baseline COSIPY v1.4 Simulation Results

Figure A1 shows a comparison between the baseline simulation results using the COSIPY v1.4 model (Sauter et al., 2020), with all default parameters and parameterisations selected, and our modified simulation. Firn temperatures differ greatly from our results and site measurements – particularly on the lower Grenzgletscher where there is a deviation in excess of 10 °C (Fig. A1i).

The deficiency for the deep firn to retain thermal energy may principally be attributed to the inability for meltwater to percolate to significant depths prior to refreezing and the thermal conductivity of sub-surface layers. Without a means to simulate preferential percolation, a process known to be highly influential in the thermal regime of a cold-infiltration firn facie (Gascon et al., 2014; Marchenko et al., 2017), percolating meltwater rarely refreezes at depths below two metres. Latent energy release therefore occurs within close proximity to the surface and is highly susceptible to rapid dissipation and loss (Fig. A1ii). This is further exacerbated by a significant overestimation in sub-surface layer thermal conductivity by the bulk-volumetric approach (Eq. 20) used in COSIPY compared to data from field and laboratory experiments (Schneebeli, 1995; Giese and Hawley, 2015) and widely used empirical formulations (Sturm et al., 1997; Calonne et al., 2019). Combined with the large positive bias in sub-surface layer density (Fig. A1iii), this leads to the entire simulated depth being strongly thermally coupled to the surface with deep firn temperatures tending towards the MAAT. It is unclear what the principal cause behind the large discrepancy between modelled and observed density is, however we noted a high sensitivity to the selection of the sub-surface re-meshing algorithm: whether the default logarithmic or adaptive Lagrangian methods of COSIPY or our fixed Lagrangian approach. There could also be a location-specific incompatibility with employing the Boone (2009) firn densification scheme, as these parameterisations have been found to have considerably variable performance at different sites (Lundin et al., 2017).

Based on these findings, we make the following main recommendations for the future development of the COSIPY model: further investigation be conducted into understanding the influence of the different re-meshing schemes on subsurface layer properties; and, given COSIPY's modular design, alternative parameterisations (eg. Calonne et al. (2019) should be added for key physical process to enhance the model's adaptability. Finally, the model is very memory intensive when executing a simulation over a moderately high spatio-temporal resolution – particularly when the user requests that subsurface variables are recorded to the output dataset. The capability to reduce the output reporting frequency would greatly reduce the size of the output data structure as it currently coupled to the input. Without access to vast computational resources, the user is otherwise forced to reduce the temporal resolution of their input meteorological data in order to prevent simulation failure.




**Figure A1. (a)** Baseline COSIPY v1.4 using the default parameter set compared against **(b)** our modified simulation results. **(i)** Modelled 20 m depth firn temperatures at the simulation end date of the 31$^{\text{st}}$ December 2023, **(ii)** modelled firn temperature-depth profile at the SP (2003-2006) and **(iii)** modelled density-depth profile compared against the CG15-1 firn core (27.09.2015) (GLAMOS, 2020; Banfi and De Michele, 2022). Spatial co-ordinates are defined by EPSG 2056 (Metric Swiss CH1903+/LV95). Topographic map source: Swisstopo (2017)



*Code and data availability.* The modified code used in this research is available at: http://doi.org/10.5281/zenodo.13361825. Meteorological
data is not available for distribution in the public domain and should therefore be requested from the respective providers.

*Author contributions.* MG modified the original source code, performed the analysis and wrote the paper. EM provided the model input data
and provided guidance on the processing steps required to extend the meteorological time series. All authors participated in the discussion
of the results.

*Competing interests.* Some authors are members of the editorial board of The Cryosphere.

*Acknowledgements.* We would like to thank ARPA Piemonte, PERMOS, DTN, and Visit Monte Rosa for providing access to their meteoro-
logical data archives. For the meteorological series of Gornergrat and Monte Rosa Plattje, these services have been provided by MeteoSwiss,
the Swiss Federal Office of Meteorology and Climatology. This work contains data/products of the Italian Air Force Weather Service. We
would also like to thank Carlo Licciulli and Josef Lier (Heidelberg University) for supplying core, borehole, and radar data. Swisstopo
and Regione Piemonte provided the digital elevation models. Past firn temperature measurements were performed within the GLAMOS
programme (Glacier Monitoring of Switzerland), financed by the Federal Office for the Environment (FOEN), MeteoSwiss, and the Swiss
Academy of Sciences (SCNAT) and maintained by the Universities of Fribourg and Zurich and ETH Zurich. The legacy data collected in the
Monte Rosa region were mainly organized and measured within the PhD thesis of Stephan Suter and were funded by the European Union
Environment and Climate Programme under ENV4-CT97-0639 and the Swiss government under BBW nr. 97.0349-1, within the framework
of the ALPCLIM EU project (Environmental and Climate records from high-elevation Alpine glaciers).


This research has been supported and partly financed by the GLAMOS program. Marcus Gastaldello received funding from the Swiss
Polar Institute (SPI) (project acronym MAGNOLIA, grant agreement SPIEG-2022-003). Enrico Mattea has been supported by the project
"Strengthening the resilience of Central Asian countries by enabling regional cooperation to assess glacio-nival systems to develop integrated
methods for sustainable development and adaptation to climate change" funded by the Global Environment Facility / United Nations Develop-
ment Programme / United Nations Educational, Scientific and cultural Organization (GEF/UNDP/UNESCO, contract no. 4500484501) and
the project "Cryospheric Observation and Modelling for Improved Adaptation in Central Asia (CROMO-ADAPT)" (contract no. 81072443)
funded by the Swiss Agency for Development and Cooperation and the University of Fribourg. Martin Hoelzle, and Horst Machguth have
been supported by the Swiss National Science Foundation SNSF (grant no. 200021 169453). Horst Machguth also acknowledges funding
by the European Research Council (ERC) under the European Union's Horizon 2020 research and innovation programme (project acronym
CASSANDRA, grant agreement No. 818994).



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
