# Peer review of "Modelling Cold Firn Evolution at Colle Gnifetti, Swiss/Italian Alps"

_EGUsphere, 2024_

## Referee Comment (RC1)

Peer review of manuscript "Modelling Cold Firn Evolution at Colle Gnifetti, Swiss/Italian Alps" by Marcus Gastaldello, Enrico Mattea, Martin Hoelzle, and Horst Machguth, submitted for publication in *The Cryosphere*.

**General comments**

This manuscript presents simulations of firn conditions at a high-elevation site in the Swiss/Italian Alps using the distributed snow model, COSIPY. The evolution of firn temperature is interrogated over the period 2003-2024, focussing on conditions in the top 30 m. Spatial patterns of firn conditions are simulated using distributed meteorological data. A continuation of previously identified trends in englacial temperature and simulated surface melt. The role of surface melt in englacial warming is discussed.

In general, the manuscript reads well, the figures are well presented, and the methods are generally well described. However, the manuscript needs to clearly articulate the additional contribution to knowledge over previous work at the same site. E.g. beyond a continuation of the warming trend observed, what new insights are provided? How does the current modelling effort differ from previous work? Do we have additional statistical confidence in the trends occurring? To this end, the manuscript may benefit from a hypothesis (or series of hypotheses) to focus the manuscript on research questions of wider significance.

I also have some concerns about the model validation that limit confidence in the model results and need addressing:

- The model tuning is somewhat ad-hoc and the validation largely occurs after the interpretation of some key model results (e.g. temperature trends, spatial patterns). A thorough validation of the model using additional data should occur before the interpretation. This could include additional comparison to near-surface and deep temperature and density profiles. A more thorough sensitivity analysis of key variables would strengthen the interpretation of key results and help justify the parameter choices.
- At present the validation of deep subsurface temperatures mixes subsurface (percolation, heat conduction) and surface (new snow density, incoming longwave and albedo) processes as well as meteorological interpolation (e.g. lapse rate). Independent validation of surface energetics (e.g. through observed surface temperature or near-surface temperature) would help give confidence in the model choices. Figure 8 indicates there is systematic variation in subsurface temperature bias with elevation and slope/aspect, yet the effect of this on the spatial patterns of trends is not discussed.
- The effect of model spin-up on the simulations results needs to be treated in a more systematic way, given the weight given to the interpretation of the large model trends at lower elevations in the manuscript. Model sensitivity to key parameters needs quantified, perhaps through a table of sensitivity runs showing model performance and changes in key results (e.g. trends) for different model configurations.
- More detail is also needed on surface energetics, specifically incoming longwave and turbulent flux calculations, especially given their connection to the development of cooling during periods of clear-skies and low wind speed that are vital to the maintenance of cold firn.

**Line comments**

4 – please clarify if melt water has infiltrated the archive already or whether it could.

9 – a 'prolongation' – is unclear what is meant here – is the previously melt rate increasing or just continuing at the same rate?

13 – 'rotation' please explain what is meant in more detail here.

13 'in the uppermost 30 metres' – the results shown in the manuscript cover the top 15-30 metres, please revise.

20-24 – either introduce the definition of polythermal in the first sentence, or define polythermal with respect to 'cold' and 'temperate' ice in the second sentence.

100 – the method used to calculate lapse rates needs introduced here or at line 113.

106 – how representative are the relative humidity measurements given the elevation difference between he sites and the target area is 600-1000 m? How sensitive is the model to potential errors in this assumption given it will control the surface energetics through portioning turbulent flux between sublimation and sensible heat.

Table 2 – would be useful to include parameterisation of incoming short and longwave radiation parameterisations in the table.

122 – are the thermistor chain data used in the analysis? If not, please remove.

159 – how was a significant snowfall event defined?

175 – what values of $c_{emission}$ were chosen?

188 – were any limits placed on the Richardson stability parameter? If not, this could introduce unrealistic cooling during periods of low wind speed. Please discuss.

Table 3 – the values for cloud transmissivity coefficient (a,b) have a large bearing on modelled LW but aren't tuned to local observations. Please discuss

Table 3 – useful to include parameters $h_{max}$ and z_interp1 and 2 (line 414) in the table as they are discussed later.

240 – Please indicate where the results of the tuning of $z_{lim}$ are shown? If this the same results as in Figure 10? If so, the top 5 metres are not shown so is hard to assess the statement. Please discuss.

259 – please refer the observations of references that indicate the overestimation of thermal conductivity.

284 – given the importance of the model spin up, further detail about the method used to construct the spin up timeseries is need, including an assessment of uncertainty.

321 – what is the justification for excluding 2003 from the analysis? how extreme was this year compared to others at other sites with longer measurement series?

Figure 5 – the shading a transparency make it hard to clearly see the changes in melt as the color scale includes a graduation in shade. Please revise.

335 – 'particularly evident at the ZS in Fig 6a' – would be useful to give years referred to as is ambiguous in figure.

Figure 7 – a map of initial temperature in 2003 would be insightful here.

Figure 7 – please add the col-temperate transition line referred to at line 454.

356 – the negative model bias at lower elevations shown in Figure 8 indicates that the early period may be too cold – how might this impact the trends shown?

359 – 'depth averaged' what depths are averaged here?

381 'depth of seasonal variation (~15m)'. Figure 6 shows the depth as between 15 and >20 m. please revise.

Figure 10 – several profiles show a poor fit to the model results (year 2000 and 2021) – perhaps this is an artefact of the seasonal variation in near-surface profile. Plots of the simulated and observed temperature profiles would give more confidence in the model results.

Also, the much flatter profile below 20 m depth indicates that model is warming too much at depth compared to observations. Integrated over the depth, this is a large difference in energy. Please discuss.

421-3 – no validation of the internal temperatures profile is made except at SP, so care must be taken when extrapolating the model confidence to unvalidated areas of different slope and aspect. Please revise.

426 – the high-resolution thermistor results should be shown here to be relevant.

454 – the transition zone is not shown in Figure 7a. please add.

476 – no statistics are shown for the correlation of the MAAT and melt – please revise.

522 – a comment on the overly cold temperature at low elevation after spin-up is needed here.

533 – 'a deviation in excess of 10 C' – not shown on figure A1i. please revise.

536 – "The deficiency for the deep firn to retain thermal energy" do the authors mean the inability of the default model scheme to simulate the warming within the firn?

**Editorial comments** (*suggested changes in italics*)

210 'Monte Moro' do you mean 'Passo del Moro'

356 'area are as of result' -> '*area are a result*'

371 'CG82-01/81' -> '*CG82-01/82*'

526 – ' thermal energy' – 'cold content' or 'thermal energy deficit' are perhaps better terms to use here.

---

## Referee Comment (RC2)

[revised manuscript text omitted]

135 $$\mathrm{SW}_\mathrm{net} \pm \mathrm{LW}_\mathrm{net} \pm Q_\mathrm{sensible} \pm Q_\mathrm{latent} \pm Q_\mathrm{subsurface} = Q_\mathrm{melt} \tag{1}$$

where $SW_{net}$ is the net shortwave flux, $LW_{net}$ is the net longwave flux, $Q_{sensible}$ and $Q_{latent}$ are the turbulent fluxes for sensible and latent exchange respectively and $Q_{subsurface}$ is the subsurface heat conduction flux. The surface temperature is physically constrained to 0 °C, therefore excess energy is apportioned to melt ($Q_{melt}$) should this condition arise.

**3.1.1 Radiative Fluxes**

140 The incident shortwave radiation for a given spatial node $(x,y)$ is modelled after Mölg et al. (2009):

$$SW_{in} = TOA\,\Lambda(x,y)\,\tau_{rg}\,\tau_{w}\,\tau_{a}\,\tau_{cl} \tag{2}$$

where TOA is the unattenuated top-of-atmosphere radiation and $\Lambda$ is a correction factor for angle of incidence and topographic shading. The coefficients of atmospheric transmissivity account for Rayleigh scattering and gaseous absorption ($\tau_{rg}$), water absorption ($\tau_{w}$) and the attenuation by aerosols ($\tau_{a}$) respectively. These are modelled after Kondratyev (1969), McDonald (1960) and Houghton (1954) respectively. We modified the parameterisation for the attenuation of cloud cover ($\tau_{cl}$) after Greuell et al. (1997) to maintain parity with the meteorological input of the EBFM:

$$\tau_{cl}(N) = 1 - aN - bN^2 \tag{3}$$

where $a = 0.233$ and $b = 0.415$ are cloud transmissivity coefficients calibrated for high altitude alpine terrain and $N$ is the fractional cloud cover. The net shortwave radiation entering the SEB is calculated using a broadband, isotropic albedo ($\alpha$):

150 $$SW_{net} = SW_{in}(1 - \alpha) - SW_{pen} \tag{4}$$

where $SW_{in}$ is the input shortwave radiation and $SW_{pen}$ represents the penetrating radiation – an apportionment of the shortwave input that bypasses the SEB and directly warms the subsurface layers. This is modelled after Bintanja and van den Broeke (1995) where the absorbed radiation at depth $z$ is calculated as:

$$SW_{pen}(z) = \lambda_{abs}\,SW_{in}(1 - \alpha)\,e^{-z\beta} \tag{5}$$

155 where $\lambda_{abs}$ is the fraction of absorbed shortwave radiation (0.8 for ice; 0.9 for snow) and $\beta$ is the extinction coefficient (2.5 for ice; 17.1 for snow).

The evolution of the albedo is modelled after Oerlemans and Knap (1998) as an exponentially decreasing function of time $t$ since the last significant snowfall event, bounded by prescribed values for fresh snow ($\alpha_{fresh}$) and firn ($\alpha_{firn}$).

160 $$\alpha(t) = \alpha_{firn} + \left[(\alpha_{fresh} - \alpha_{firn}) \cdot e^{-\frac{t}{t^*}}\right] \tag{6}$$

where $t^*$ is the characteristic decay timescale parameter. We modified this by adding the enhancement of Bougamont et al. (2005) that enables both a faster decay on a melting surface and slower metamorphism in cold conditions by introducing a dependence on surface temperature:

$$
t^*(T_s) = \begin{cases} t^*_{\mathrm{wet}}, & T_{\mathrm{s}} = 0\,^{\circ}\mathrm{C} \\ t^*_{\mathrm{dry}} + K\left[\max(T_{\mathrm{s}}, T_{\mathrm{max},\,t^*})\right], & T_{\mathrm{s}} < 0\,^{\circ}\mathrm{C} \end{cases}
\tag{7}
$$

165    where $t^*_{\mathrm{wet}}$ and $t^*_{\mathrm{dry}}$ are the decay timescales (d) for a melting and dry surface respectively, $K$ is a calibration parameter and $T_{\mathrm{max},\,t^*}$ is a temperature threshold for the decay timescale adjustment.

Net longwave radiation is calculated in accordance with the Stefan-Boltzmann law for grey body emission:

$$
\mathrm{LW}_{\mathrm{net}} = \mathrm{LW}_{\mathrm{in}} - \varepsilon_{\mathrm{s}}\sigma T_{\mathrm{s}}^4
\tag{8}
$$

170    where $\varepsilon_{\mathrm{s}}$ is the surface emissivity and $\sigma$ is the Stefan-Boltzmann constant. In lieu of any input longwave radiation data at the site, the model parameterises this flux by substituting the air temperature and atmospheric emissivity ($\varepsilon_{\mathrm{atm}}$) into the Stefan-Boltzmann law (Konzelmann et al., 1994).

$$
\varepsilon_{\mathrm{atm}} = \varepsilon_{\mathrm{cs}}(1 - N^2) + \varepsilon_{\mathrm{clouds}}N^2
\tag{9}
$$

$$
\varepsilon_{\mathrm{cs}} = 0.23 + c_{\mathrm{emission}}\left(\frac{e_{\mathrm{a}}}{T_{\mathrm{a}}}\right)^{0.125}
\tag{10}
$$

175    where $\varepsilon_{\mathrm{cs}}$ and $\varepsilon_{\mathrm{clouds}}$ are the clear sky and cloud emissivities respectively and $c_{\mathrm{emission}}$ is a calibration parameter.

**3.1.2   Turbulent Fluxes**

The turbulent fluxes are calculated according to a bulk aerodynamic approach after Foken (2008); Stull (1988):

$$
Q_{\mathrm{sensible}} = \rho_{\mathrm{a}}c_{\mathrm{p,a}}C_{\mathrm{h}}V_a(T_a - T_{\mathrm{s}})
\tag{11}
$$

$$
Q_{\mathrm{latent}} = \rho_{\mathrm{a}}L_{\mathrm{s,v}}C_{\mathrm{h}}V_a(q_a - q_{\mathrm{s}})
\tag{12}
$$

[revised manuscript text omitted]

---

## Author Comment (AC2)

**REVIEWER 1 COMMENTS (Anonymous)**

| ID | Comment | Response | Action |
|---|---|---|---|
| #1 (General) | The manuscript needs to clearly articulate the additional contribution to knowledge over previous work at the same site. E.g. beyond a continuation of the warming trend observed, what new insights are provided? How does the current modelling effort differ from previous work? Do we have additional statistical confidence in the trends occurring? To this end, the manuscript may benefit from a hypothesis (or series of hypotheses) to focus the manuscript on research questions of wider significance. | The original intentions of this research project were to implement a different energy balance firn model for comparison with the EBFM used by Mattea et al. (2021); extend the meteorological forcing to the present day (2019-2023); and improve upon the methods of the original study.

Initial attempts to do this were greatly hindered by several critical bugs and limitations in the original version 1.4 of the COSIPY model. These problems were rectified by this research and formed a major contribution to the release of the latest version 2.0.

At Colle Gnifetti, the original study by Mattea et al. (2021) established a good baseline but had considerable limitations. Improving the model spin-up and the firn warming trends has been a key focus. Since the draft submission, I have further refined the spin-up to be based on a long-term lapse rate adjusted Mean Monthly Air Temperature (MMAT) from Jungfraujoch. This makes the simulation of firn temperatures much more accurate as the initial conditions are more representative. The previous model of Mattea et al. (2021) also did not include the process of subsurface melting and used a less accurate parameterisation of thermal conductivity – quite influential for the thermal | Research developments are now more explicitly stated in the manuscript (eg. improvements to model spin-up, modelling of firn temperature gradients, contributions to COSIPY v2.0 development etc.) |

| | | | |
|---|---|---|---|
| | | regime of firn. Nor was there any validation of modelled firn density. The updated Figure 10 shows considerable improvement in the development of the thermal gradient, as it much more accurately matches observed values. This figure also showcases the important observed englacial warming at Colle Gnifetti (Figure 10), that is as yet unpublished in a scientific journal. In general, I think that the draft did not articulate well the considerable developments that have taken place since the prior research – I have now stated them more explicitly. | |
| #2 (General) | The model tuning is somewhat ad-hoc and the validation largely occurs after the interpretation of some key model results (e.g. temperature trends, spatial patterns). A thorough validation of the model using additional data should occur before the interpretation. This could include additional comparison to near-surface and deep temperature and density profiles. A more thorough sensitivity analysis of key variables would strengthen the interpretation of key results and help justify the parameter choices. | Generally, it is extremely difficult to constrain many of these model parameters and at Colle Gnifetti, being an exposed high-altitude location, data is very scarce. I would generally prefer to refrain from placing the model validation prior to the presentation of the results as it disrupts the natural flow of the manuscript. I feel they are better placed afterwards to lead into the discussion. | Sensitivity analysis been added to the manuscript (Appendix B) and temperature & density profiles (observed & modelled) comparing to all observational data is included in the supplementary material. |
| #3 (General) | At present the validation of deep subsurface temperatures mixes subsurface (percolation, heat conduction) and surface (new snow density, incoming longwave and albedo) processes as well as meteorological interpolation (e.g. lapse rate). Independent validation of surface energetics (e.g. through observed surface temperature or near- | There is unfortunately limited data for the validation of surface energy exchanges – the exposed position of the site makes it extremely difficult to maintain monitoring equipment. Thus, we are limited to the short-term, temporary weather stations established at Seserjoch and Lysjoch. All parameters are | Better explanation of how model parameters and surface energy exchanges were calibrated to observations of local |

| | | | |
|---|---|---|---|
| | surface temperature) would help give confidence in the model choices. Figure 8 indicates there is systematic variation in subsurface temperature bias with elevation and slope/aspect, yet the effect of this on the spatial patterns of trends is not discussed. | calibrated to the available data from the site with the exception of the preferential percolation depth ($z_{lim}$) and new snow density ($\rho_0$) – that have been tuned to fit the thermistor profiles and firn density cores respectively.

Topographic influences on firn temperatures were not discussed in great detail because it was mostly viewed as intuitive to well-established knowledge of the influence of elevation on air temperature and aspect/slope on solar insolation, however this has now been added. | temporary monitoring stations has been added. Additional discussion of the influence of topography on the spatial variation of firn temperatures. |
| #4 (General) | The effect of model spin-up on the simulations results needs to be treated in a more systematic way, given the weight given to the interpretation of the large model trends at lower elevations in the manuscript. Model sensitivity to key parameters needs quantified, perhaps through a table of sensitivity runs showing model performance and changes in key results (e.g. trends) for different model configurations. | Fully agreed. Therefore, I have developed a more accurate model spin-up from 1939 using meteo data from Jungfraujoch adjusted via a monthly variable lapse rate to the Colle Gnifetti altitude. This initialises the firn at the model start date to conditions representative of 2003 and a significant improvement has been observed in the results. There are still limitations with this approach, but it is considered to be far better method than that used in the first version of the manuscript.

The results of the sensitivity study are now included in the manuscript. | Results are updated with the new model spin-up. This has a limited effect on surface interactions, mainly influencing deep firn temperatures – notably decreasing englacial warming at depth and making the model correlate better with observed profiles (Figure 10). The firn temperature gradient is now also much more accurate.

Sensitivity study results included as Appendix B. |

| | | | |
|---|---|---|---|
| #5 (General) | More detail is also needed on surface energetics, specifically incoming longwave and turbulent flux calculations, especially given their connection to the development of cooling during periods of clear-skies and low wind speed that are vital to the maintenance of cold firn. | (See response to comment #2 and #3) Agreed – these are important factors although as previously expressed, data scarcity and the remoteness of this location does make validation difficult. | Further discussion on this subject has been added in the manuscript. |
| #6 (Line 4) | *'Infiltrating meltwater compromises this vital archive'* - please clarify if melt water has infiltrated the archive already or whether it could. | This sentence a general statement about the consequences of firn warming in the Alps – rather than a specific indication of the status at Colle Gnifetti.

At Colle Gnifetti, it is difficult to conclusively state the extent to which melt is influencing firn stratigraphy, especially as there is great spatial variation, however the most recent cores extracted from the Signalkuppe flank in 2021 were not significantly influenced. However, while there is evidence of melt suggesting the region is no longer a recrystallisation firn facie, there is currently no evidence it percolates to significant depth. | None – this section of the text precedes the introduction to the study site and at Colle Gnifetti, the huge spatial variability and lack of observational data means it is difficult to make conclusive statements about this. |
| #7 (Line 13) | *'prolongation'* – is unclear what is meant here – is the previously melt rate increasing or just continuing at the same rate? | The word prolongation is used here to express that the trend has simply continued with time at around the same rate – regression values being recalculated with the additional data. There is too much interannual variation to infer whether the rate has increased in the last 5 years. | Substitute with 'continuation' to clarify that the rate remains unchanged. |
| #8 (Line 13) | *'rotation'* - please explain what is meant in more detail here. | This is referring to the fact that there is an inversion in the temperature gradient at this depth, that has been increasing and so has 'rotated'. It would perhaps be clearer to use better terminology and state that a temperature inversion has developed. | Sentence revised for clarity to express that an inversion in the gradient has developed and its strengthening. |

| | | |
|---|---|---|
| #9 (Line 13) | *'In the uppermost 30 metres'* – the results shown in the manuscript cover the top 15-30 metres, please revise. | Agreed – this simplification ignores the strong fluctuations near the surface. | Sentence revised to improve accuracy – eg. *'In the 17-30 m depth range'* |
| #10 (Line 20/24) | *'Leading these glaciers to attain a hybrid or polythermal regime'* - either introduce the definition of polythermal in the first sentence or define polythermal with respect to 'cold' and 'temperate' ice in the second sentence. | This sentence has been revised as the initial clause *'The majority of glaciers within the European Alps exist in a fully temperate state'*, is also a bit of a generalisation that is not fully accurate. | Sentence revised for clarity. |
| #11 (Line 100) | the method used to calculate lapse rates needs introduced here or at line 113. | The temperature lapse rate is that shown in Figure 2 – it is calculated based on the temperature difference between stations in the region at an hourly resolution . For pressure, it's the barometric equation. | The methods used to calculate temperature/pressure lapse rates are now included. |
| #12 (Line 106) | how representative are the relative humidity measurements given the elevation difference between the sites and the target area is 600-1000 m? How sensitive is the model to potential errors in this assumption given it will control the surface energetics through portioning turbulent flux between sublimation and sensible heat. | This assumption was made based on an analysis of the variability of relative humidity at all stations in the study area (Table 2). At stations above 3,000 m, values remained similar with the variable no longer being particularly correlated to elevation as at lower altitudes. | Discussion of this assumption has been added – and more generally the estimation of unmeasured variables is included in the sensitivity study. |
| #13 (Table 2) | would be useful to include parameterisation of incoming short and longwave radiation parameterisations in the table. | Comment accepted. | Shortwave and longwave parameterisations have been added to Table 2 and stated in the main body. |
| #14 (Line 122) | are the thermistor chain data used in the analysis? If not, please remove. | Yes, they are mentioned in line 425 of the discussion and now included in the supplementary material. | None |
| #15 (Line 159) | how was a significant snowfall event defined? | This is a model parameter that was set to 0.1 mm. Due to the extremely low accumulation rate on the Signalkuppe slope, this value must be kept extremely low otherwise significant | This parameter has also been added to Table 3. |

| | | material is unaccounted for in this area of the spatial domain.

Note that the albedo is reset to the albedo of the second layer, should the first one melt entirely. | |
|---|---|---|---|
| #16
(Line 175) | what values of c_emission were chosen? | A value of 0.42 was selected based on Mattea et al.'s (2021) calibration of the temporary station at Seserjoch (Suter et al.,2004). It is mentioned in Table 3. | Parameter values are now also stated in the main body of the text. |
| #17
(Line 188) | were any limits placed on the Richardson stability parameter? If not, this could introduce unrealistic cooling during periods of low wind speed. Please discuss. | The method of Essery and Etchevers (2004) was used as originally described in their paper. The Richardson stability parameter ($\Psi_{Ri}$) is limited between 0 and 1 according to Eq. 15). Therefore, even if the Richardson number is extremely high at low wind speeds, there is no unrealistically high turbulent energy flux. | None. |
| #18
(Table 3) | the values for cloud transmissivity coefficient (a,b) have a large bearing on modelled LW but aren't tuned to local observations. Please discuss | These were calibrated to the global radiation data from the AWS Capanna Margherita – this should have been stated in Table 3. | Missing calibration method to added to Table 3. |
| #19
(Table 3) | useful to include parameters h_max and z_interp1 and 2 (line 414) in the table as they are discussed later. | Comment accepted. | Parameters added to Table 3. |
| #20
(Line 240) | Please indicate where the results of the tuning of zlim are shown? If this the same results as in Figure 10? If so, the top 5 metres are not shown so is hard to assess the statement. Please discuss. | The results of the tuning of $z_{lim}$ were not presented in the draft manuscript but now are shown in the sensitivity study. | Sensitivity study with $z_{lim}$ tuning results added. |
| #21
(Line 259) | please refer the observations of references that indicate the overestimation of thermal conductivity. | Calonne et al. (2019) Fig. 1 shows an evaluation of thermal conductivity against many commonly used parameterisations. I have annotated the bulk volumetric approach below as it is not shown in this figure since the method is rarely used due to it being quite a | Reference to Calonne et al. (2019) added. |

| | | simplistic approach that heavily overestimates thermal conductivity.  | |
|---|---|---|---|
| #22 (Line 284) | given the importance of the model spin up, further detail about the method used to construct the spin up timeseries is need, including an assessment of uncertainty. | Comment accepted (see response to comment #4). Further information has been provided about the model spin-up and its limitations – especially given as how it is shown to be rather important. | Further discussion on the model spin up methodology has been added – especially following recent updates (see response to comment #4) |
| #23 (Line 321) | what is the justification for excluding 2003 from the analysis? how extreme was this year compared to others at other sites with longer measurement series? | Comment accepted. Justification has been provided based on comparative data from the cryosphere in Switzerland compared to other years. | Justification/references added. |
| #24 (Figure 5) | the shading a transparency make it hard to clearly see the changes in melt as the color scale includes a graduation in shade. Please revise. | The purpose of using transparency was that the topography can still be visible to the viewer and a sequential colour map was chosen due to the nature of melt as a sequential variable. I have now increased the hue range of the colour to better distinguish different bands and superimposed a transparent map – rather than using transparency in the shading. | Figure has been revised to ensure clarity in viewing the results. Small visual adjustments have also been made to other figures. |

| #25 (Line 335) | *'particularly evident at the ZS in Fig 6a'* – would be useful to give years referred to as is ambiguous in figure. | Comment accepted. | High melt years such as 2003, 2015 and 2022 to now explicitly stated. |
|---|---|---|---|
| #26 (Figure 7) | a map of initial temperature in 2003 would be insightful here. | Agreed – I was just trying to only select the most important figures to avoid making the manuscript too lengthy. A figure is now provided in the supplementary material and the raw data will also be provided as a NetCDF on a Zenodo. | Additional figure added to the supplementary material and raw data accessible via NetCDF. |
| #27 (Figure 8) | please add the col-temperate transition line referred to at line 454. | Figure revised to demarcate model nodes that are temperature and included a boundary transition line. I also presume this refers to Figure 7. | Figure revised. |
| #28 (Line 356) | the negative model bias at lower elevations shown in Figure 8 indicates that the early period may be too cold – how might this impact the trends shown? | Comment accepted. | Further discussion on the influence on initial firn temperatures on reported trends added. |
| #29 (Line 359) | 'depth averaged' what depths are averaged here? | The entire depth range of the measurement (i.e. if the measurement is between 4m and 24 m, the bias is calculated against the simulation in this range. | Text revised to express how the measurement is averaged over the corresponding depth of measurement. |
| #30 (Line 381) | 'depth of seasonal variation (~15m)'. Figure 6 shows the depth as between 15 and >20 m. please revise. | Figure 6 shows the simulation results, so the simulated ZAA is not necessarily equivalent. The value of 15 metres is based on observations only – however a more conservative value would be 17 m. Note that there are additional measurements in 2019, 2021 and 2022 not shown in Figure 10 that make this more visible, however including all measurements heavily clutters the figure as they are all concentrated together. | Figure 10 updated with line showing the observed ZAA line at 17 metre depth. |

| | | |
|---|---|---|
| #31
(Figure 10) | several profiles show a poor fit to the model results (year 2000 and 2021) – perhaps this is an artefact of the seasonal variation in near-surface profile. Plots of the simulated and observed temperature profiles would give more confidence in the model results. Also, the much flatter profile below 20 m depth indicates that model is warming too much at depth compared to observations. Integrated over the depth, this is a large difference in energy. Please discuss. | Fully agree with these comments. The results using the new model spin-up (see response to comment #4), show far better agreement at depth with a more accurate reduced warming at depth. | Results updated using new model spin up.

Temperature & density profiles (observed & modelled) added to the supplementary material. |
| #32
(Line 421) | no validation of the internal temperatures profile is made except at SP, so care must be taken when extrapolating the model confidence to unvalidated areas of different slope and aspect. Please revise. | (See previous comment) – I do have access to these, but I had not included them in the manuscript draft. These are now included. | Temperature & density profiles (observed & modelled) added to the supplementary material. |
| #33
(Line 426) | the high-resolution thermistor results should be shown here to be relevant. | Unfortunately, our snow depth sensor on-site was destroyed meaning that we are unable to accurately determine the depth of the thermistors relative to the surface. A few of the upper thermistors also melted out of the snowpack.

Essentially the data provides an insight into the nature of the importance of preferential percolation at the site, but it is difficult to analyse further. Our measuring equipment has now been repaired and hopefully future research will not suffer from these issues. | The measured values from these high resolution thermistors are now included in the supplementary material with a depth uncertainty of ±50 cm, as they do at least demonstrate the importance of modelling deep percolation and refreezing. |
| #34
(Line 454) | the transition zone is not shown in Figure 7a. please add. | Comment accepted. See response to comment #27 | Figure revised. |
| #35
(Line 476) | no statistics are shown for the correlation of the MAAT and melt – please revise. | Comment accepted. | Additional statistics included for this correlation. |

| | | | |
|---|---|---|---|
| #36 (Line 522) | a comment on the overly cold temperature at low elevation after spin-up is needed here. | Comment accepted. | Discussion regarding how firn temperature bias is likely strongly influenced by initial temperatures being too cold has been added. |
| #37 (Line 533) | *'a deviation in excess of 10 C'* – not shown on figure A1i. please revise. | This deviation is shown by comparison of firn temperatures in Figures A1(a)(i) and A1(b)(i) – the spatial maps on the leftmost column. | None |
| #38 (Line 536) | *'The deficiency for the deep firn to retain thermal energy'* do the authors mean the inability of the default model scheme to simulate the warming within the firn? | Yes – this could be expressed clearer. | Sentence adjusted for clarity. |
| #39 (Line 210) | 'Monte Moro' do you mean 'Passo del Moro' | Comment accepted. Consistent naming will be used. | Correction made. |
| #40 (Line 356) | 'area are as of result' -> 'area are a result' | Comment accepted. | Correction made. |
| #41 (Line 371) | *'CG82-01/81'* -> 'CG82-01/82' | Comment accepted. | Correction made. |
| #42 (Line 526) | ' thermal energy' – 'cold content' or 'thermal energy deficit' are perhaps better terms to use here | Comment accepted. | Alternative suggestion now used. |

**REVIEWER 2 COMMENTS (Adrien Gilbert)**

| ID | Comment | Response | Action |
|---|---|---|---|
| #1 (General) | The study would be more interesting if the sensitivity of the results to the different parameters were clearly analysed and reported. In particular, I miss the influence of the modification you made compared to Mattea et al. (2021). How sensitive is the result to penetrating radiation? To surface roughness? To thermal conductivity? To percolation depth? This exercise has already been partially done in the appendix of Mattea et al. (2021) by single parameter modification (2 values). This could be done in more detail in this study by better exploring the parameter space and the associated modelled temperature change. | A sensitivity study was carried out and included in the study. | Sensitivity study results are now included as Appendix B and briefly discussed. |
| #2 (General) | You mention a high-resolution thermistor chain at SP that allows to identify a significant 4m-deep temperature increase in 21 days. It would be interesting to see how the model compares to this record in detail to potentially identify the origin of the model bias. This is especially interesting because summer 2022 has the most intense high altitude melt ever recorded ... | Unfortunately, our snow depth sensor on-site was destroyed meaning that we are unable to accurately determine the depth of the thermistors relative to the surface. A few of the upper thermistors also melted out of the snowpack.

Essentially the data provides an insight into the nature of the importance of preferential percolation at the site, but it is difficult to analyse further. Our measuring equipment has now been repaired and hopefully future research will not suffer from these issues. | Unfortunately, it is difficult to fully exploit this data fully due to equipment failure and the high depth uncertainty.

However, I have now included these measured temperatures in the supplementary material with a depth uncertainty of ±50 cm, as they do at least demonstrate the importance of modelling deep |

| | | | percolation & refreezing onsite. |
|---|---|---|---|
| #3 (General) | The authors highlight, as in Mattea et al. (2021), the strong influence of the imposed percolation depth and suggest that a more physical approach would be better, but nothing has been done. Implementing an alternative, more physical approach (even simple), for water percolation would be an improvement/modification that would better justify a new paper. You could also use your high-resolution thermistor chain at SP to validate the percolation scheme. | The original intentions of this research project were to implement a different energy balance firn model for comparison with the EBFM used by Mattea et al. (2021); extend the meteorological forcing to the present day (2019-2023); and improve upon the methods of the original study.

However, the COSIPY model in its version 1.4 rendition was found to have critical bugs in its refreezing module and its calculation of the surface temperature, as well as many other smaller issues. A large proportion of this work has gone towards the development of this model which has recently been re-released at version 2.0 with the incorporation of many of the amendments made in this study.

In response to the first reviewers' comments, a more accurate model spin-up from 1939 using meteorological data from Jungfraujoch adjusted via a lapse rate has also been added. This has greatly improved the simulation of temperature gradients and englacial warming compared to the method originally employed by Mattea et al. In general, that has been a key focus of our new study, as the original study did not investigate firn temperatures in great detail. As shown in Figure 9, their evolution over | The developments made in this research are now more explicitly expressed in the manuscript. The contribution this work has made to the COSIPY model development is now also mentioned.

While the percolation scheme remains unchanged, other major limitations in the original study have been rectified, so that firn temperatures and their evolution over time is much more accurate. |

| | | time in Mattea et al. is not very representative.

This paper also showcases the important observed englacial warming at Colle Gnifetti (Figure 10), that is as yet unpublished in a scientific journal.

Implementing a physically based percolation scheme is considered beyond the scope of this paper but will certainly be the foremost priority for my future research at Colle Gnifetti. It is also worth stating, that simply doing this would be heavily constrained by a lack of data on snow microstructure and stratigraphy – something I intend to collect in the future to ensure parameters can be calibrated. Resolving the issues with the sonic ranger will also assist here. | |
|---|---|---|---|
| #4
(General) | The model tends to have a cold bias compared to the observation. It would be interesting to identify the origin of this bias. A detailed comparison of model and data at a selected site could help. I think it is possible to show if the bias comes from heat conduction, vertical advection, missing energy input within the snowpack (radiation penetration?), or from too cold surface temperature from the SEB model? Or something else? | I think it is difficult to conclusively state whether the model has a cold bias based on the heavily weighted distribution of validation profiles on shaded slopes. There is also the issue that some past measurements at Colle Gnifetti have been measured shortly after steam drilling and have a high uncertainty in estimating their equilibrated temperatures. This is one of the main reasons why I state in the discussion that it is very important to establish a better spatially distributed set of reliable validation profiles. This is currently part of my future research plans, and I hope this can be achieved in the | Further discussion added regarding the cold bias of the model – principally on the area of the lower Grenzgletscher slope. |

| | | | |
|---|---|---|---|
| | | coming years. Regardless, there certainly could be more discussion about model bias in the paper. | |
| #5 (Line 1) | *'Firn'* – Ice or glaciers. Cold firn in itself has limited paleo-climatic interest | Comment accepted. | Amended to 'cold firn and ice' |
| #6 (Line 4) | *'Physical transition between the different thermal regimes'* - It is not clear what do you mean by "physical transition" | This simply meant the transition between a cold firn facie and temperate firn facie. Perhaps the word 'physical' is superfluous. | Sentence amended. |
| #7 (Line 5) | *'Modified version'* - what is modified? which physical processes are now included? This should be clearly stated in the abstract if this is the novelty compared to Mattea et al. 2021 | The modifications to the COSIPY model are extensive including many parametrisation changes (see Table 2), critical bug fixes and optimisations. Mattea et al. (2021) also used a different model – the Energy Balance Firn Model (EBFM) of van Pelt et al. (2012). While in principle I agree with your comment, I don't believe it's possible to state such a multitude of changes in a concise enough manner to fit into the abstract of this paper. | Abstract adjusted to be more specific – however I do want to keep it short and concise, so it isn't possible to contain all these specifics. |
| #8 (Line 8) | *'Capanna Margherita'* – 'weather station' | Comment accepted. | Wording adjusted. |
| #9 (Line 13) | *'Rotation in the temperature gradient'* – I guess you mean inversion of the temperature gradient | Inversion would certainly be the more accurate terminology to explain the direction of the temperature gradient, but I also wanted to describe how it has slowly 'rotated' from a non-inverted / steady state in the 1980s to an inverted state today. | Sentence revised to express that a temperature gradient inversion is present in the 17 – 30m depth range that is strengthening over time. |
| #10 (Line 16) | *'Model parameterisation'* - which one? | Principally the preferential percolation depth ($z\_lim$), snow density ($\rho\_0$) and albedo ($\alpha$). | See response to comment #7 |
| #11 (Line 18) | *'Model spin-up have a major influence'* – how? because of different initial thermal state? Be more precise in the two last sentence of the abstract. | Yes, but also the initial state of the firn and its intrinsic subsurface properties (eg. density, thermal conductivity, heat capacity etc.) | Abstract reworded to be less vague – however it is not possible to be too specific here. |

| | | (See responses to comments #7) | See response to comment #7 |
|---|---|---|---|
| #12 (Line 24) | *(Suter and Hoelzle, 2002; Blatter and Hutter, 1991)* – Miss the reference from the Mont Blanc area: Gilbert et al. 2015 / Gilbert and Vincent (2013). You could add Lüthi and Funk (2001). The reference to Blatter and Hutter is irrelevant here as you speak about the European Alps. Lüthi, M. P. and Funk, M.: Modelling heat flow in a cold, high-altitude glacier: interpretation of measurements from Colle Gnifetti, Swiss Alps, Journal of Glaciology, 47, 314–324, https://doi.org/10.3189/172756501781832223, 2001. Gilbert, A., Vincent, C., Gagliardini, O., Krug, J., and Berthier, E.: Assessment of thermal change in cold avalanching glaciers in relation to climate warming, Geophysical Research Letters, 42, 6382–6390, https://doi.org/10.1002/2015GL064838, 2015. Gilbert, A. and Vincent, C.: Atmospheric temperature changes over the 20th century at very high elevations in the European Alps from englacial temperatures, Geophysical Research Letters, 40, 2102–2108, https://doi.org/10.1002/grl.50401, 2013. | Comment accepted. | Additional references added. |
| #14 (Line 43) | *(Vincent et al., 2017)* – No future evolution modelling in this may be. More relevant to cite: Gilbert, A., Vincent, C., Six, D., Wagnon, P., Piard, L., and Ginot, P.: Modelling near-surface firn temperature in a cold accumulation zone (Col du Dôme, French Alps): from a physical to a semi-parameterized approach, The Cryosphere, 8, 689–703, https://doi.org/10.5194/tc-8-689-2014, 2014. | Comment accepted. | Reference changed to that suggested. |
| #15 (Line 44) | *(Hoelzle et al., 2011)* – Miss reference to: Gilbert, A. and Vincent, C.: Atmospheric temperature changes over the 20th century at very high elevations in the European Alps from englacial temperatures, Geophysical Research | Comment accepted. | Additional reference added. |

| | | | |
|---|---|---|---|
| | Letters, 40, 2102–2108, https://doi.org/10.1002/grl.50401, 2013. | | |
| #16 (Line 71) | *'Constrained'* – limited? | I think in this context both words have similar meaning but constrained implies more a restriction in scope rather than just size. | None |
| #17 (Line 75) | *'Thus, snow accumulation has a strong summer bias...'* - very unclear sentence | Comment accepted. | Sentence reworded for improved clarity. |
| #18 (Line 79) | *'Lateral mass transference processes'* – this is not English. You mean 'transfer' | Comment accepted. | Correction made. |
| #19 (Line 84) | *'Alternative coupled energy balance and multi-layer subsurface firn model to CG'* – you could add a sentence to briefly describe what is new compared with Mattea et al. (2021) | Comment accepted. | This paragraph has been amended to better reflect the advancements of this study and changes from Mattea et al. (2021) |
| #20 (Line 86) | *'The nature of both simulated and observed firn warming'* – what does it mean? | Firn temperatures simulated by the models and firn temperatures observed from on-site measurements. | None |
| #21 (Line 92?) | ??? – there is a problem is this sentence | It is unclear as to what this comment refers to – missing link in the PDF file. | – |
| #22 (Line 102) | *'Fractional cloud cover'* – why cloud cover is needed if you directly measure incoming short and long wave radiation? | Longwave radiation is not measured at the Capanna Margherita; therefore, it must be parameterised using fractional cloud cover. While observed shortwave radiation is accurate for the Capanna Margherita, in order to adjust it for the topographic inclination (slope/aspect) of other grid cells it is necessary to calculate the TOA radiation. This also preserves parity with the radiative flux input of Mattea et al. (2021) | None. |
| #23 (Line 112) | *'Whilst'* - while | British English – has the same meaning as 'while', but more commonly used in the British dialect when expressing contrast. | None. |

| | | | |
|---|---|---|---|
| #24 (Line 104) | *'The remaining meteorological variables (windspeed, relative humidity and cloud cover) are assumed constant with elevation.'* - What about incoming short-wave radiation? You could add here a description of how it is spatialised or refer to section 3.1.1.You could also say that incoming LW are computed from humidity, cloud cover and air temperature. | The radiative fluxes are heavily parameterised and variable across the spatial domain, according to equations (2) and (8). I consider them to be 'derived values' of the model and not input meteorological data, therefore they are introduced in Section 3. Whilst I could put a forward reference to section 3.1.1., I would consider this bad writing practice. | None. |
| #25 (Section 3) | You need to specify the unit of all variables and parameters. Some are missing. | Comment accepted. | Missing specification of variables/ parameters added. |
| #26 (Line 155) | From where these values come from? The unit are missing. I assume you use m^-1. Subsurface temperatures are very sensitive to this parameter so its value should be more discussed.For example, Gilbert et al. (2014) found that the appropriate value to model subsurface temperature at col du Dôme (very similar setup) is Beta= 40 m-1 (could be cited). Fukami et al. (1985) also suggests higher value than 17 m-1.Gilbert, A., Vincent, C., Six, D., Wagnon, P., Piard, L., and Ginot, P.: Modelling near-surface firn temperature in a cold accumulation zone (Col du Dôme, French Alps): from a physical to a semi-parameterized approach, The Cryosphere, 8, 689–703, https://doi.org/10.5194/tc-8-689-2014, 2014.Fukami, H., Kojima, K., and Aburakawa, H.: The Extinction and Absorption of Solar Radiation Within a Snow Cover, Annals of Glaciology, 6, 118–122, https://doi.org/10.3189/1985AoG6-1-118-122, 1985. | These are the fixed (hard-coded) parameters in the COSIPY model. I had previously not considered modifying them being unaware this may have such significant implications – thank you for bringing this to my attention. | Suggested value for snow extinction coefficient adopted (40 m-1), given site similarity and limited local observational data to define this new parameter.

Parameter added to Table 3 (extinction coefficient for ice not changed but not particularly important in a deep firn zone like CG). |
| #27 (Line 170) | *'In lieu of'* – English? Why not use 'instead' | British English – both words/phrases have identical meaning – the former is perhaps more formal. | None. |

| | | | |
|---|---|---|---|
| #28 (Line 170) | '$T_a$' – Do you use the elevation dependent temperature computed from the lapse rate? | Yes, for any given model node, air temperature and pressure are adjusted according to their elevation differences from Capanna Margherita using variable lapse rates (see line 111). Specifically, (although unmentioned in the manuscript) the local lapse rate is calculated based on the hourly temperature difference between Capanna Margherita and the other regional stations and the pressure is adjusted using the barometric equation. | The use of lapse rates to determine nodal air temperatures is already described in the manuscript. However, the sentence has been adjusted to more specifically explain how they are calculated. |
| #29 (Eq. 12) | $q_s$ - how do you estimate qs? As a function of Ts assuming saturation? | Yes, the specific humidity is calculated based on the parameterisation of Sonntag (1994) that determines the saturated vapour pressure as a function of temperature. This should have been mentioned in the draft version of the manuscript. | Reference to the parameterisation of Sonntag (1994) added with a brief explanation. |
| #30 (Line 186) | *'and μ is the surface roughness.'* – give unit | Correction accepted. | Unit (m) added. |
| #31 (Line 195) | 'μ' – This is usually an important parameter that needs to be specifically calibrated. Its value should be discussed through a more detailed literature review. For example, in similar setup, Gilbert et al. (2014) found 0.004 m.You could also look at Brock, B. W., Willis, I. C., and Sharp, M. J.: Measurement and parameterization of aerodynamic roughness length variations at Haut Glacier d'Arolla, Switzerland, Journal of Glaciology, 52, 281–297, https://doi.org/10.3189/172756506781828746, 2006. | The default COSIPY model uses the parameterisation of Moelg et al. (2012) to calculate surface roughness as a linearly increasing function of time t since the last snowfall event. The problem with using this is that the snow surface is heavily eroded by wind scouring at Colle Gnifetti. Generally, all model parameters were tested in a thorough sensitivity study, and surface roughness was found to be one of the least sensitive parameters – it is therefore not included in the new figure in Appendix B. | No change. Using a constant based on observational data on wind profiles from Suter et al. (2004) at Seserjoch is considered the best approach for this particular site. |

| #32 (Line 213) | *'By default,'* – What do you mean ?? | This is referring to the default approach used by the COSIPY model. | None. |
|---|---|---|---|
| #33 (Line 214) | *'However, we reverted to using a constant value'* – This sentence is unclear. you mean that you use a constant surface density? | Yes, the density of new snow added to the uppermost layer is always 250 kg m-3. However, as these layers are constituted from multiple precipitation/snowfall events to reach their 10 cm thickness, their effective density is typically considerably higher than this value. Perhaps this should be reworded as the 'fresh snow density'. | Phrase reworded to improve clarity. |
| #34 (Line 214) | *'Layers can translate vertically in the grid...'* – I think it should be clarified that the Lagrangian scheme allows to take into account vertical heat advection and its dependence to snow accumulation. This is an important process which is not really mentioned. | Comment accepted. | Processes of vertical advection also now detailed in the description of the Lagrangian scheme. |
| #35 (Line 226) | I thought you use density dependent conductivity using Calonne et al. (2019)? (Table 2) | I modified the thermal conductivity parameterisation to Calonne et al. (2019) but the model default is what is stated here.

The reason these paragraphs are structured in this way is because I wanted to express the default parameterisations in the COSIPY model and then express the modifications we made to the original model. | None

(See response to comment #40) |
| #36 (Line 232) | *'We supplemented this with...'* – you could explain that water is introduced in the firn within in given thickness instead of all in the first layer. | I am unsure of the meaning of this comment, but I think it relates to comment #38. Here I am explaining the default approach of the model and then introducing our changes. | None. (See response to comment #38) |
| #37 (Line 233) | *'Due to our assessment that the default bucket approach...'* – which assessment? | This was more of a manner of speaking regarding the research, but I agree this is a bit of a sweeping generalisation. Essentially, employing the baseline bucket approach of COSIPY leads to firn temperatures being | Sentence reworded to briefly express how the basic bucket approach was found to be unrepresentative as it |

| | | much colder than their observed values, as meltwater refreezing all occurs in the uppermost metre of the firn. But I didn't want to introduce too much discussion into the methodology section of the paper. | lacked a means to simulate preferential percolation - with a reference to Appendix A provided. |
|---|---|---|---|
| #38 (Eq. 21) | So, there is no temperature dependent criteria ??? what happen if the 4 first meter are temperate? I see from the next paragraph that you do a bucket scheme at end. So, this PDF is only use for the initial liquid water input (distributed instead of all in the first layer). This should be clear from the beginning of the section. | No – employing a more sophisticated percolation scheme is out of scope but will be the direct focus of my future research. (See response to comment #3)

Yes – this is why I used the word 'supplemented' to try and express that the 'bucket' approach is not completely removed. | Paragraph adjusted to ensure it is clear to the reader that the Marchenko preferential percolation scheme adds onto the bucket scheme and does not completely replace it. |
| #39 (Line 254) | *'Effective density'* – how this relate to Phi_ice, Phi_air …. ? rho=Phi_air ? | The properties of all subsurface layers are derived as a weighted sum of their fractional composition (see Line 226 / Eq.20).

$\rho_{eff} = \rho_{ice}\,\phi_{ice} + \rho_{water}\,\phi_{water} + \rho_{air}\,\phi_{air}$ | Equation 20 has been modified to express that this is what the 'effective' (eff) value represents. |
| #40 (Line 258) | It is confusing to first say that Eq. 20 is used to estimate the thermal conductivity to finally say you use something else. | I agree that this may add a bit of confusion, however I wanted to express the differences between the default parameterisations / methods of COSIPY and our modifications (particularly relevant to Appendix A and those interested in using the model). It is therefore necessary in my opinion to mention both. | Small adjustments have been made to the methodology to ensure it is clear when the original and altered model parameterisations are being referred to , however the general structure remains unchanged. |
| #41 (Line 273) | *'Furthers this by…'* – I am not sure this sentence is correct (English) | Meaning 'enhances this', 'improves upon this' etc. | None. |

| | | | |
|---|---|---|---|
| #42 (Line 299) | *'Has to compensate with a greater expenditure of energy through melt'* – why not just saying that this extra energy produces more melt? There is nothing to compensate… I guess you refer to Eq (1) but it makes more sense to say that more energy is available to melt. The word "expenditure" is probably unnecessary complicated. | Agreed – this wording needlessly overcomplicates the sentence. | Sentence simplified. |
| #43 (Line 309) | *'The magnitude of longwave emission is physically limited by the surface temperature constraints'* - This is a very important point which is at the origin of the energy excess absorbed by the firn during melting. The 0°C surface temperature condition limits LW emission but also the latent and sensible flux through qs and Ts being limited. You could look Figure 5 of Gilbert et (2014) about this.Gilbert, A., Vincent, C., Six, D., Wagnon, P., Piard, L., and Ginot, P.: Modelling near-surface firn temperature in a cold accumulation zone (Col du Dôme, French Alps): from a physical to a semi-parameterized approach, The Cryosphere, 8, 689–703, https://doi.org/10.5194/tc-8-689-2014, 2014. | Comment accepted. | I have added to the discussion regarding the insights of this paper and added the reference. |
| #44 (Line 332) | *'Whilst'* - while | British English – has the same meaning as 'while', but more commonly used in the British dialect when expressing contrast. | None. |
| #45 (Line 340) | *'Thermal diffusion subsequently…'* – and advection which can be the dominant process if significant snow accumulation. | This an important point to add, however I will specify it as 'vertical' advection to avoid confusion with lateral advection (not modelled by 1-D COSIPY). | Suggestion added. |
| #46 (Caption of Figure 10) | *'Bias corrected simulated englacial warming'* – it is not clear to what refer this bias corrected simulation and why it provides a range. | The bias correction refers to the fact that the model overestimates the firn/ice temperature to be + 0.4 °C warmer than that which is observed. The concept behind this figure is to show the temperature variation between the start of the simulation (following the spin-up) in January 2003 and | Caption reworded to explain that this is the model results corrected for its temperature bias. Also now stated in the text main body. |

| | | its end in December 2023 – thus the grey enclosed region shows the modelled englacial warming. The purpose of subtracting the bias is therefore to allow the reader / viewer to directly compare the model to the observations. | |
|---|---|---|---|
| #47 (Line 395) | *'Our results are not considered fully validatory given the operational...'* – maybe reformulate and be more precise with what you mean | Agreed – this sentence is a bit general and could be more specific. | Sentence reworded to be more specific. |
| #48 (Line 403) | *'Furthermore, given that the surface model was calibrated and produces a SEB in accordance with the energy fluxes measured at Seserjoch'* - Not clear how this validating something. It sounds very qualitatively since you refer to flux measurement from more than 20 years ago. | There certainly are limitations using this observational data, but this is all we have available, and we consider it to still be a valid source of information. It provides an insight to the relative magnitude and contribution of different fluxes on the energy balance. Whilst there is certainly interannual variation and a trend of increasing melt, there is limited temporal variation in these fluxes. | Added a brief statement of the limitations of heavily depending on this data source. |
| #49 (Line 425) | *'Our high-resolution thermistor chains at the SP provide insight into the importance of this process, showing a remarkable increase of 7.74 °C at 4 m depth over the course of 20 days during the summer melt season of 2022'* – why don't you show these data and how it compares to the model. It can be a good way to identify what are the model limitation. | (See response to comment #2) | Measurements from thermistor chains added to the supplementary material, albeit with a depth uncertainty of ±50 cm. Further use of this data is difficult due to this high uncertainty. |
| #50 (Line 427) | *'The default bucket approach of COSIPY was found to result in an underestimation of modelled firn temperatures by as much as 10 °C on the lower Grenzgletscher slope'* - I am surprised by this result. Are you sure you compare two simulations with the same amount of simulated melting? | Yes, this is the result of a sensitivity study with a simulation run with all parameters/parameterisations controlled except switching the percolation scheme. | No direct changes, but I have now included the sensitivity of the z_lim value in Appendix B's new sensitivity study |

| | | | |
|---|---|---|---|
| #51
(Line 441) | *'The dimensionality of COSIPY…'* – I don't think it is true as long as all the water refreeze in the first meters. I also don't see the aim of this paragraph. Is it a justification to not do any physical approach for percolation because it is too complicated? Given that this study brings little new things compare to Mattea et al. (2021). An improvement of the percolation scheme would have been something that could justify a new paper …. | The purpose of this paragraph was simply to acknowledge the current limitations of the model and to suggest that including a means to simulate lateral flow would also be an important development for modelling percolation on steep mountain slopes. | None.

See response to comment #3 regarding percolation scheme. |
| #52
(Line 455) | *The tuning of key model parameters based on conditions at the CG saddle* – We don't really know which parameter you are tuning, how and based on what? | The parameter selection choices are stated in Table 3, however this could also be more explicitly stated in the main body of the text. The preferential percolation depth ($z\_lim$) is the tuned to the firn temperature profiles and the fresh snow density ($\rho\_0$) is tuned to the firn density profiles. Other parameters are set based on previous research from the site with references provided (eg. Mattea et al. (2021), Luthi and Funk (2001) and Suter et al. (2004). | More information about parameter tuning added to the main body of text rather than just being stated in Table 3. |
| #53
(Line 458) | *'Akin'* –? | British English – meaning 'of similar nature or character'. A synonym of 'like' that is more formal. | None |
| #54
(Line 488) | *'In actuality'* – ? *English?* | British English – used to emphasize something different is actually the case in contrast to what has been previously said. | None |
| #55
(Line 503) | *'This was interpreted by Vincent et al. (2020) to be a consequence of the formation of low-permeability ice layers that limit the percolation depth of meltwater and consequently the retainment of latent heat release from refreezing.'* - The author mainly explains the cooling from the climate warming hiatus between 1998 and 2015 since direct modelling using observed climate data reproduces the observed cooling.The author said:"This | Comment accepted. | Sentence reworded to better reflect the discussion of this study and state that is a suggestion of Vincent et al. (2020) to explain the different warming trend in the Mont Blanc region |

| | | | |
|---|---|---|---|
| | means that the 2017 stabilization observed at a depth of 40 m is a signature of an air temperature warming rate slowdown observed in the Lyon-Bron climatic data between 1998 and 2015 and well known on a global scale as "the global warming hiatus" (Meehl et al., 2014)."But it is true that the effect of ice layer limiting water percolation is mentioned in the discussion as it can potentially limit the warming effect, but Vincent et al. do not provide any evidence of it. You could reformulate the way you refer to this study. | | – not that there is actual evidence of significant ice lens formation. |
| #56 (Line 511) | *'Modifications to several of the model's parameterisations for key physical processes' -* you could list here these modifications. It is an important part of the paper. | The default parameterisations of the COSIPY v1.4 model are listed, in contrast to those used in our model and the EBFM study of Mattea et al. (2021), in Table 2. However, I don't think it is good writing practice to place in-text references in a conclusion, nor can I easily summarise the large number of changes between the two models here. | None – the changes are simply too extensive to include in the conclusion in detail. |